# Hydrate-melt electrolyte design for aqueous aluminium-bromine batteries with enhanced energy-power merits

Xingyuan Chu[1,4], Jingwei Du[1,4], Jiaxu Zhang[1], Xiaodong Li [1,2], Xiaohui Liu[1], Yongkang Wang [3], Johannes Hunger [3], Ahiud Morag[1], Jinxin Liu [2], Quanquan Guo[1,2], Dongqi Li [1], Yu Han [3], Mischa Bonn [3], Xinliang Feng [1,2] ✉ & Minghao Yu [1,2] ✉

Aluminium-based aqueous batteries hold promises for next-generation sustainable and large-scale energy storage due to the favorable metrics of Al and water-based electrolytes. However, the performance of current aluminium-based aqueous batteries falls significantly below theoretical expectations, with a critical bottleneck of realizing cathodes with high areal capacities. Herein, we present a hydrate-melt electrolyte design utilizing cost-effective $AlCl_3$ and organic halide salts, which enables the demonstration of aqueous Al-Br batteries with enhanced energy-power characteristics. The optimal electrolyte features suppressed water activity and loosely bound halogen anions, attributed to its unique electrolyte structure, where the majority of water molecules engage in robust ion solvation (>98% as suggested by simulations) and halogen anions reside in the outer solvation sheath of cations. These distinctive features ensure good compatibility of the electrolyte with the reversible $Br^-/Br^0/Br^+$ conversion, enabling cathodes with a high areal capacity of 5 mAh cm$^{-2}$. Besides, the electrolyte allows for Zn-Al alloying/de-alloying with minimal polarization (around 100 mV at 5 mA cm$^{-2}$) and a smooth alloy surface. The assembled Al-Br cell delivers an energy density (267 Wh L$^{-1}$, based on the volume of anode, cathode and separator) comparable to commercial Li-ion batteries and a substantial power density (1069 W L$^{-1}$) approaching electrochemical capacitors.

Rechargeable aqueous batteries utilizing non-flammable and green water-based electrolytes exhibit desirable safety and sustainability, representing a promising frontier of next-generation battery technologies for large-scale and stationary applications[1,2]. Aqueous Zn batteries, utilizing Zn metal as anodes (negative electrodes) and mild Zn salt solutions as electrolytes, have emerged as a notable example, making significant strides in both academic research and preliminary industrial technology transfer[3]. Compared with Zn, Al offers

advantageous metrics in terms of abundance (8.2% vs. 0.0075% in Earth's crust) and specific capacity (2980 mAh g$^{-1}$ vs. 820 mAh g$^{-1}$, 8046 mAh cm$^{-3}$ vs. 5855 mAh cm$^{-3}$)[4], which thereby motivates the exploration of Al-based aqueous batteries (AABs). Recent efforts have yielded promising results in reversible Al stripping/plating in aqueous solutions through strategies like electrolyte optimization[5,6], anode engineering[7-10], and interphase construction[11]. These advancements in the anode pave the way for developing high-performance AABs. On the

[1]Center for Advancing Electronics Dresden (cfaed) & Faculty of Chemistry and Food Chemistry, Technische Universität Dresden, Dresden, Germany. [2]Max Planck Institute of Microstructure Physics, Halle (Saale), Germany. [3]Department of Molecular Spectroscopy, Max Planck Institute for Polymer Research, Mainz, Germany. [4]These authors contributed equally: Xingyuan Chu, Jingwei Du. ✉e-mail: xinliang.feng@tu-dresden.de; minghao.yu@tu-dresden.de

other hand, large-capacity and high-voltage cathodes (positive electrodes) are imperative for assembling AAB devices with sufficient energy densities. In many reported studies, manganese-based oxides have been employed as cathode materials, demonstrating a large gravimetric capacity (~500 mAh g$^{-1}$) and an average discharge voltage of 1.5 V[7,9–12]. Other alternative cathode materials have also been explored, including intercalation-type (e.g., Prussian blue analogs)[10], conversion-type (e.g., sulfur)[10], and organic (e.g., polyaniline) materials[5]. Despite these efforts, AAB cathodes are often evaluated with low mass loadings (<2 mg cm$^{-2}$), resulting in impractical areal capacity (<1 mAh cm$^{-2}$) and energy density (<1 mWh cm$^{-2}$). This limitation, in turn, leads to an unrealistic N/P ratio (i.e., the capacity ratio between the anode and the cathode, normally > 10) and a drastically reduced energy density of the cell.

Enhancing the areal capacity of AAB cathodes poses significant challenges primarily because the triply charged Al$^{3+}$ acts as the dominant charge carrier. In conventional cathode materials encompassing intercalation-type, conversion-type, and organic compounds, the electrochemical charge storage processes are often governed by solid-state ion diffusion. The substantial charge density of trivalent Al$^{3+}$ (364 C mm$^{-3}$) induces strong repulsive interactions with host cathodes[13], leading to sluggish ion diffusion and consequently hindered electrochemical reaction kinetics. This limitation is further evidenced by the unsatisfactory rate capability observed in conventional cathodes. For instance, the specific capacity of an Al$_x$MnO$_2$ electrode with a low mass loading of 1 mg cm$^{-2}$ was noted to decrease from 0.46 mAh cm$^{-2}$ at 0.1 mA cm$^{-2}$ to 0.10 mAh cm$^{-2}$ at 3.0 mA cm$^{-2}$ [7]. In contrast to conventional cathodes, the anionic Br redox reaction (Br$^-$/Br$^0$) can undergo a liquid (Br$^-$)-to-quasi solid (Br$_2$) conversion with the assistance of a confining host and Br-species dissolution-inhibiting electrolyte additives (e.g., salts with bulky organic cations)[14]. This distinctive Br$^-$/Br$^0$ conversion overcomes the hindrance associated with solid-state Al$^{3+}$ diffusion during the charge storage process and exhibits high reaction kinetics that could potentially allow for a practical areal capacity akin to metal stripping/plating reactions. Moreover, an additional conversion step (Br$^0$/Br$^+$) tends to be initiated with the participation of Cl$^-$ through interhalogen reactions, forming Br-Cl compounds[15]. Thus, the overall Br$^-$/Br$^0$/Br$^+$ conversion presents two redox steps at 1.05 and 1.30 V vs. standard hydrogen electrode (SHE) with a theoretical capacity of 670 mAh g$_{Br}^{-1}$, representing a promising cathode chemistry for constructing high-energy AABs. However, such

aqueous Al-Br batteries remain unexplored due to the critical challenges in developing a suitable aqueous electrolyte that simultaneously supports the Br$^-$/Br$^0$/Br$^+$ conversion and reversible Al-based anode chemistry.

To tackle the above challenges, this study reports a hydrate-melt electrolyte design employing cost-effective AlCl$_3$ together with organic bromide salts, which enables the realization of aqueous Al-Br battery devices with enhanced energy-power merits (Fig. 1). As the optimal electrolyte, 3.25 m (moles of a solute per kilogram of a solvent) AlCl$_3$ + 1 m 1-butyl-1-methylpyrrolidinium bromide (PY14Br) exhibits strongly ion-associated water molecules and loosely bound halogen anions. Ab initio molecular dynamics (AIMD) simulations suggest that >98% of water molecules in the electrolyte engage in robust ion solvation, encompassing the two solvation sheaths of Al$^{3+}$ and one solvation sheath of other ions. This analysis underscores the hydrate-melt nature of the electrolyte, where the vast majority of water molecules are strongly coordinated with ions, leaving only a minimal amount of free water molecules[16,17]. This unique hydrate-melt feature significantly suppresses water activity, as evidenced by various characterizations, and expands the electrochemical stability of the electrolyte up to 1.5 V vs. Ag/AgCl. Consequently, the electrolyte allows reversible Br$^-$/Br$^0$/Br$^+$ conversion, enabling electrodes with an areal capacity of 5 mAh cm$^{-2}$. Moreover, the hydrate-melt electrolyte demonstrates good compatibility with Zn-Al alloy anodes, allowing for electrochemical Zn-Al alloying/de-alloying with low voltage polarization (around 100 mV at 5 mA cm$^{-2}$) and a smooth alloy surface. Combining the Br$^-$/Br$^0$/Br$^+$ conversion cathode with the Zn-Al alloy anode yields aqueous Al-Br cells with an average discharge voltage of 1.7 V. Considering the large areal capacity and rapid electrochemical kinetics of both electrodes, the cell delivers a volumetric energy density (267 Wh L$^{-1}$), well comparable to the commercial Li-ion batteries and a large volumetric power density (1069 W L$^{-1}$) approaching electrochemical capacitors.

## Results
### Hydrate-melt electrolytes for Br$^-$/Br$^0$/Br$^+$ conversion

To formulate an aqueous electrolyte fitting into Br$^-$/Br$^0$/Br$^+$ conversion, cheap AlCl$_3$ was chosen as the solute given two considerations: 1) AlCl$_3$ allows for a high-concentration solution, wherein massive Al$^{3+}$ cations with strong solvation capability can significantly suppress the H$_2$O activity, thus enhancing the anodic stability of the electrolyte; 2) free Cl$^-$ anions play a crucial role in promoting Br$^0$/Br$^+$ conversion. The first consideration was justified by the dielectric relaxation spectra of AlCl$_3$ solutions with different concentrations (i.e., 0.10 m, 1.00 m, and 3.25 m). Typically, the dielectric relaxation spectra detect the response of the electrolytes under a time-dependent small-amplitude electric field, reflecting the macroscopic dipole moment of the solutions[18,19]. Compared with pure water solvent, the dielectric constant (i.e., the permittivity below 100 GHz) drastically decreases along with the increase of AlCl$_3$ concentration (Supplementary Fig. 1). In the dielectric loss spectra (Fig. 2a), the amplitude of the main peak (~20 GHz), primarily associated with bulk water in solutions, becomes very low as the AlCl$_3$ concentration increases from 0 (pure water) to 3.25 m. This result indicates that most water molecules in 3.25 m AlCl$_3$ are tightly 'bonded' with ions and thereby cannot freely rotate under the external electric field. Moreover, the signatures of anion-cation pairs, which typically contribute to lower frequency (~2.5 GHz) components of the dielectric loss, are negligible in all AlCl$_3$ solutions, indicating a weak Al$^{3+}$-Cl$^-$ interaction[20].

Raman spectra of electrolyte solutions further confirmed the suppressed water activity (Supplementary Fig. 2). Compared with pure water, 3.25 m AlCl$_3$ presents an obviously diminished intermolecular O-H peak at 3220 cm$^{-1}$, reflecting a reduction in free water behavior[21,22]. This pronounced suppression of water activity contributes to an extended electrochemically stable potential window. The anodic stability of all AlCl$_3$ solutions was assessed utilizing a three-electrode

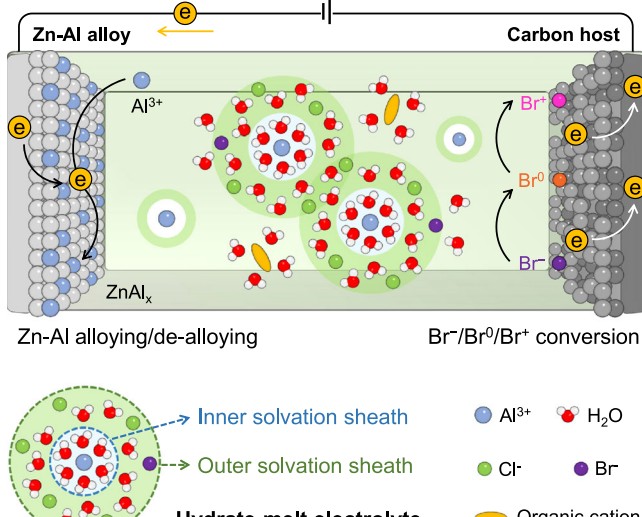

**Fig. 1 | Schematic of an Al-Br battery enabled by a hydrate-melt electrolyte.** The hydrate-melt electrolyte is featured by the predominant presence of water molecules engaged in ion solvation sheath and loosely bound halogen anions.

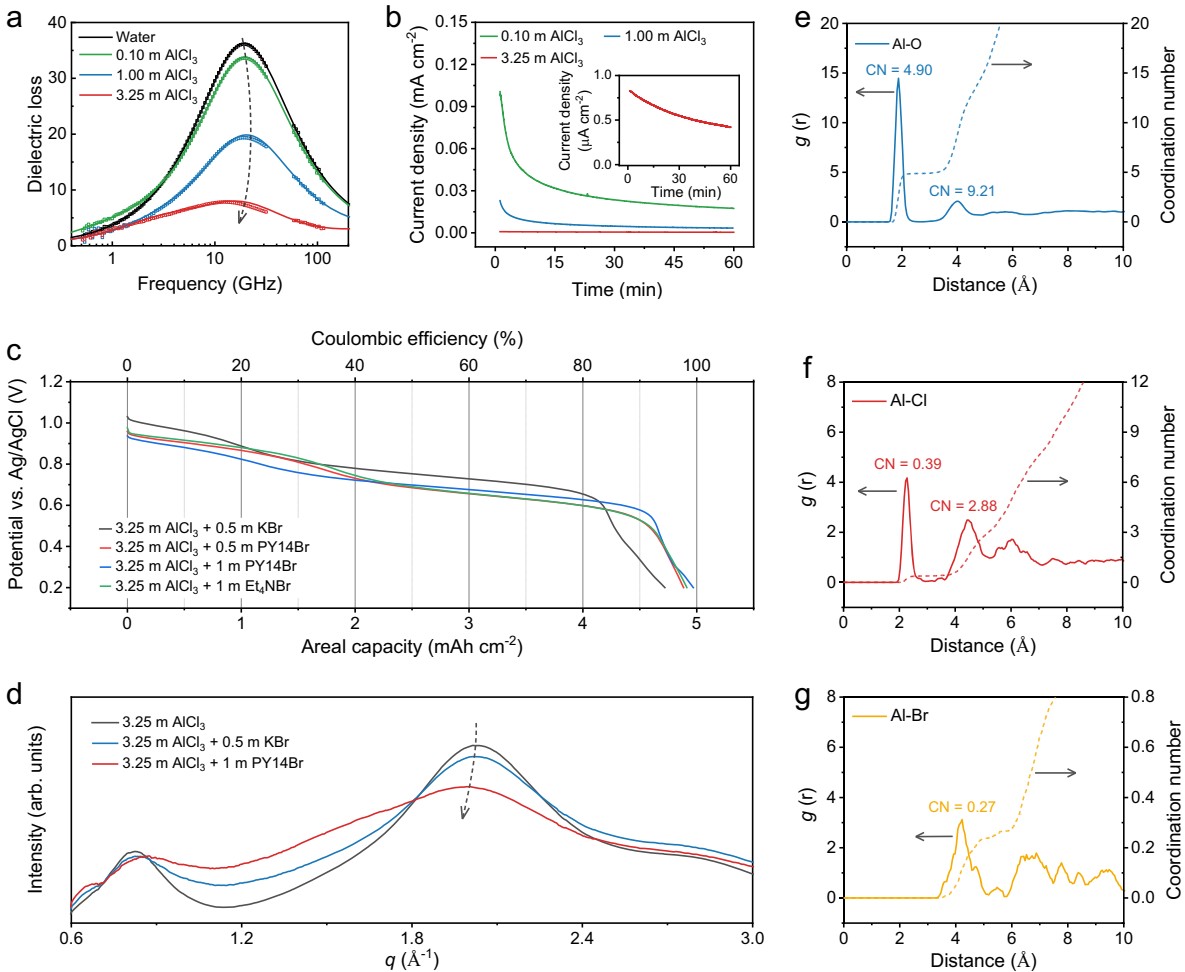

**Fig. 2 | Hydrate-melt electrolyte design and properties. a** Dielectric loss spectra of aqueous AlCl₃ solutions with varying concentrations, with corrections applied for contributions from the *d.c.*-conductivity. Symbols correspond to experimental data and solid lines show the fitting curves. **b** Potential floating test of the Ti electrode in AlCl₃ aqueous electrolytes at 1.5 V vs. Ag/AgCl. **c** Galvanostatic discharge profiles of the AC electrode at 5 mA cm⁻² in different electrolytes. **d** WAXS patterns of 3.25 m AlCl₃, 3.25 m AlCl₃ + 0.5 m KBr, and 3.25 m AlCl₃ + 1 m PY14Br. RDFs (solid lines) and integral curves (dashed lines) of Al atoms extracted from AIMD simulations of 3.25 m AlCl₃ + 1 m PY14Br, including **e** Al–O_water, **f** Al–Cl, and **g** Al–Br. The RDF data generated in this study are provided in Supplementary Data 3. Source data are provided as a Source Data file.

electrochemical cell with a Ti rod, an Ag/AgCl electrode, and an activated carbon (AC) electrode as the working, reference, and counter electrodes, respectively. As expected, both potential floating (Supplementary Fig. 3) and linear sweep voltammetry (LSV) tests (Supplementary Fig. 4) manifest the substantially boosted anodic stability of 3.25 m AlCl₃. It is notable that 3.25 m AlCl₃ shows a negligible leakage current density of 0.6 µA cm⁻² at a high potential of 1.5 V vs. Ag/AgCl (ca. 1.7 V vs. SHE) (Fig. 2b). This high anodic stability establishes 3.25 m AlCl₃ as a promising base electrolyte for further design to accommodate the Br⁰/Br⁺ conversion (1.3 V vs. SHE).

With 3.25 m AlCl₃ confirmed as a promising base electrolyte, we next introduced bromide salts into 3.25 m AlCl₃ for Br⁻/Br⁰/Br⁺ conversion. Three different bromide salts were assessed, including KBr, PY14Br, and tetraethylammonium bromide (Et₄NBr). Of note, the introduction of organic cations does not compromise the anodic stability of the electrolyte (Supplementary Fig. 4). All the obtained electrolytes remain transparent and free of precipitates after three months of storage at room temperature (Supplementary Fig. 5), demonstrating their good stability. The addition of bromide salts causes slight changes in the viscosities and ionic conductivities of these electrolytes compared to 3.25 m AlCl₃ (Supplementary Fig. 6 and Supplementary Table 1). Br⁻/Br⁰/Br⁺ conversion was evaluated in a three-electrode

Swagelok cell with an AC electrode (15 mg cm⁻², 278 µm in thickness) as the hosting electrode, an over-capacity AC electrode as the counter electrode, and the Ag/AgCl reference electrode. We screened the bromide salts using a galvanostatic charge-discharge (GCD) measurement at 5 mA cm⁻² with 5 mAh cm⁻² as the charge cut-off and a potential of 0.2 V vs. Ag/AgCl as the discharge cut-off. In the charge profiles (Supplementary Fig. 7), two plateaus were observed for all the bromide salt-containing electrolytes, reflecting the two-step Br⁻/Br⁰/Br⁺ conversion. Likely, all the electrolytes display two discharge plateaus (Fig. 2c), but with varying areal capacities. In detail, 3.25 m AlCl₃ + 0.5 m KBr exhibits the lowest areal capacity of 4.72 mAh cm⁻² with a coulombic efficiency of 94.4%. By contrast, organic bromide salts can considerably boost the coulombic efficiency to 98.1%, 99.6%, and 98.6% for 3.25 m AlCl₃ + 0.5 m PY14Br, 3.25 m AlCl₃ + 1 m PY14Br, and 3.25 m AlCl₃ + 1 m Et₄NBr, respectively. Notably, the capacity predominantly arises from the Br⁻/Br⁰/Br⁺ conversion, and the capacitive charge storage of the AC electrode accounts for only 0.28 mAh cm⁻² (Supplementary Fig. 8). We also evaluated the self-discharge issue of the charged electrode in all electrolytes after 24-h open-circuit standing. The capacity retention of 3.25 m AlCl₃ + 0.5 m PY14Br, 3.25 m AlCl₃ + 1 m PY14Br, and 3.25 m AlCl₃ + 1 m Et₄NBr achieved 89.1%, 95.2%, and 92.8%, respectively, considerably surpassing 65.6%

achieved with 3.25 m AlCl$_3$ + 0.5 m KBr (Supplementary Fig. 9). These results unveil that both 3.25 m AlCl$_3$ + 1 m PY14Br and 3.25 m AlCl$_3$ + 1 m Et$_4$NBr can initiate Br$^-$/Br$^0$/Br$^+$ conversion with high coulombic efficiencies, with 3.25 m AlCl$_3$ + 1 m PY14Br exhibiting slightly superior performance metrics. Moreover, these electrolytes offer a substantial cost advantage over the commonly reported Al(OTF)$_3$-based electrolytes used in aqueous Al batteries (Supplementary Tables 2, 3)[9,11,12].

To understand the micro-species in the optimal 3.25 m AlCl$_3$ + 1 m PY14Br electrolyte, we compared its[23] Al nuclear magnetic resonance (NMR) spectrum with those of 3.25 m AlCl$_3$ and 3.25 m AlCl$_3$ + 0.5 m KBr (Supplementary Fig. 10). All three electrolytes present a dominant peak at 0 ppm, which corresponds to water-solvated Al$^{3+}$ (i.e., Al(H$_2$O)$_6$$^{3+}$). No characteristic peaks associated with halogen-Al$^{3+}$ pairing were detected[24]. Moreover, Fig. 2d displays the wide-angle X-ray scattering (WAXS) patterns of the three electrolytes with a scattering vector ($q$) range of 0.6 - 3 Å$^{-1}$, covering the characteristic distance ($d$) from 10.5 to 2.1 Å based on Eq. (1). In 3.25 m AlCl$_3$, two peaks centered at $q$ = 0.8 Å$^{-1}$ and $q$ = 2 Å$^{-1}$ were detected, referring to the distance between neighboring solvated Al$^{3+}$ cations ($d$ = 7.8 Å) and the intermolecule distance of water ($d$ = 3.1 Å), respectively[25,26]. The addition of 0.5 m KBr or 1 m PY14Br causes negligible intensity change in the peak at $q$ = 0.8 Å$^{-1}$, implying non-disruption in the solvation configuration of Al$^{3+}$ in 3.25 m AlCl$_3$ + 0.5 m KBr and 3.25 m AlCl$_3$ + 1 m PY14Br. A notable change was observed for the peak at $q$ = 2 Å$^{-1}$ in 3.25 m AlCl$_3$ + 1 m PY14Br, which, compared with 3.25 m AlCl$_3$ and 3.25 m AlCl$_3$ + 0.5 m KBr, shows pronounced peak broadening towards lower $q$ value. This observation indicates the presence of less bulky water molecule aggregates in 3.25 m AlCl$_3$ + 1 m PY14Br compared with 3.25 m AlCl$_3$ and 3.25 m AlCl$_3$ + 0.5 m KBr. The relatively more disrupted hydrogen-bonding network in 3.25 m AlCl$_3$ + 1 m PY14Br contributes to suppressing water activity. A similar conclusion can be drawn by the comparison of the dielectric loss spectra for the three electrolytes (Supplementary Fig. 11). The spectrum of 3.25 m AlCl$_3$ + 1 m PY14Br exhibits a notably suppressed peak associated with bulk water aggregates at 20 GHz, in contrast to those of 3.25 m AlCl$_3$ and 3.25 m AlCl$_3$ + 0.5 m KBr.

$$d = 2\pi/q \tag{1}$$

Ab initio molecular dynamics (AIMD) simulations were further conducted to obtain molecule-level insights into the three electrolytes (Supplementary Fig. 12 and Supplementary Data 1–3). The high accuracy of our AIMD simulations was validated by deriving WAXS spectrum using Debye's scattering equation[23], which aligns with the experimental spectra (Supplementary Fig. 13). The derived radial distribution functions (RDFs) from the AIMD simulations disclose the neighboring Al−Al distance range of 6 - 10 Å in all three electrolytes (Supplementary Figs. 14−16), consistent with the WAXS analysis. These electrolytes exhibit similar structures, where the majority of water molecules engage in robust ion solvation, encompassing two solvation sheaths around Al$^{3+}$ and one solvation sheath around other ions. Halogen anions predominantly reside in the outer solvation sheath of Al$^{3+}$. Specifically, in the RDF analysis of Al in 3.25 m AlCl$_3$ + 1 m PY14Br, two strong Al-O$_{water}$ peaks are identified at 1.88 Å and 4.03 Å, indicating the presence of two solvation sheaths around Al$^{3+}$ with the coordination number (CN) of 4.90 and 9.21 (Fig. 2e). Of note, the water solvation energy of Al$^{3+}$ in the outer solvation sheath even exceeds that of Li$^+$ in the inner solvation sheath (Supplementary Fig. 17 and Supplementary Data 4−6), evidencing the strong capability of Al$^{3+}$ in limiting water activity.

In 3.25 m AlCl$_3$ + 1 m PY14Br, dominant Cl$^-$ (79.5 %) and Br$^-$ (75.0 %) anions are revealed to reside in the outer solvation sheath of Al$^{3+}$ with an Al−Cl distance of 4.5 Å (Fig. 2f) and an Al−Br distance of 4.2 Å (Fig. 2g). Only a small fraction of Cl$^-$ anions (13.3% estimated from the CN of 0.39) are observed in the inner solvation sheath of Al$^{3+}$ with an

Al−Cl distance of 2.3 Å. The representative Al$^{3+}$ solvation structure in 3.25 m AlCl$_3$ + 1 m PY14Br also illustrates the location of Cl$^-$ anions in the outer solvation sheath (Supplementary Fig. 18), which contrasts with the direct anion-cation pairing detected in ionic-liquid Al battery electrolytes or highly concentrated ZnCl$_2$ aqueous electrolyte[27,28]. These loosely bound Br$^-$ and Cl$^-$ anions are expected to facilitate fast interfacial charge transfer during Br$^-$/Br$^0$/Br$^+$ conversion by avoiding kinetics-limiting anion-cation dissociation, which is supported by density functional theory calculations (Supplementary Fig. 19). Additionally, statistical analysis of water molecules involved in strong solvation reveals minimal free water (i.e., water not engaged in strong solvation, including the two solvation sheaths of Al$^{3+}$ and one solvation sheath of other ions) in 3.25 m AlCl$_3$ (6.1%), 3.25 m AlCl$_3$ + 0.5 m KBr (5.4%), and 3.25 m AlCl$_3$ + 1 m PY14Br (1.8%), underscoring the hydrate-melt nature of all three electrolytes (Supplementary Table 4). The lowest free water content in 3.25 m AlCl$_3$ + 1 m PY14Br is attributed to the addition of PY14Br, where both Br$^-$ and PY14$^+$ significantly contribute to water molecule association (Supplementary Fig. 16).

## Br$^-$/Br$^0$/Br$^+$ conversion mechanism

We further sought to elucidate the Br conversion mechanism in 3.25 m AlCl$_3$ + 1 m PY14Br. As shown in the cyclic voltammetric (CV) profile (Fig. 3a), the conversion reaction shows two pairs of redox peaks at 0.6 - 0.7 V vs. Ag/AgCl and 0.9 - 1.0 V vs. Ag/AgCl, which align with the two-step charge/discharge profiles. The redox peaks at the lower potential are attributed to the Br$^-$/Br$^0$ conversion based on the aligned potential, while the redox reaction at the higher potential corresponds to the Br$^0$/Br$^+$ conversion[15]. Furthermore, the Br$^-$/Br$^0$/Br$^+$ conversion was validated by the Br K-edge X-ray absorption near-edge structure (XANES) spectra of the AC electrode at different depths of charge (DOC = 0%, 50%, and 100%) during a charge cycle at 5 mA cm$^{-2}$ and 5 mAh cm$^{-2}$. Standard Br$_2$-adsorbed AC and PY14Br were measured as references for analysis. As shown in Fig. 3b, the electrode at 0% DOC shows a nearly identical spectrum as the PY14Br reference, confirming the presence of only Br$^-$. At 50% DOC, the spectrum of the AC electrode undergoes an apparent negative shift and is almost overlapped with that of Br$_2$. This observation unveils Br$^-$/Br$^0$ as the first conversion step. Moreover, the electrode at 100% DOC exhibits no edge shift compared with the electrode at 50% DOC, but the edge peak at 13,473 eV is drastically intensified. Typically, this peak is associated with the intra-atomic 1$s$-to-4$p$ electron transition of Br, and its intensity is indicative of the hole density of Br 4$p$ orbitals[15]. The high peak intensity of the electrode at 100% DOC verifies the increased hole density in Br 4$p$ orbital, namely the presence of Br$^+$.

Additionally, in-situ Raman spectra (Fig. 3c) were collected in a home-made two-electrode electrochemical cell (Supplementary Fig. 20) to understand the interhalogen species during the conversion reaction. At the fully discharged state, the electrode shows almost no peaks, as the Br$^-$ is monatomic ion in the electrolyte, showing no Raman activity. Along charging for 1.25 mAh cm$^{-2}$, the electrode starts to show characteristic peaks associated with Br$_3$$^-$ (at 160 cm$^{-1}$) and Br$_{2n+1}$$^-$ (at 205 cm$^{-1}$ and 252 cm$^{-1}$)[29]. Moreover, charging the electrode for 5 mAh cm$^{-2}$ induced the presence of a new peak at 273 cm$^{-1}$, a characteristic peak for the symmetric stretching vibration of BrCl$_2$$^-$[30]. These results verify the conversion process from Br$^-$ to Br$_{2n+1}$$^-$/BrCl$_2$$^-$, without involving sluggish solid-state Al$^{3+}$ diffusion. In 3.25 m AlCl$_3$ + 1 m PY14Br, the PY14$^+$ cations can enhance the Br$^-$/Br$^0$/Br$^+$ conversion reversibility by inhibiting the dissolution of active polyhalide species (e.g., Br$_{2n+1}$$^-$ and BrCl$_2$$^-$) from the AC electrode into the electrolyte (Supplementary Fig. 21). These organic PY14$^+$ cations can electrostatically associate with anionic polyhalide species, forming water-insoluble phase that prevent their dissolution into the aqueous electrolyte (Supplementary Fig. 22). In this regard, organic cations with a stronger binding affinity toward anionic polyhalide species than toward water molecules are more likely to

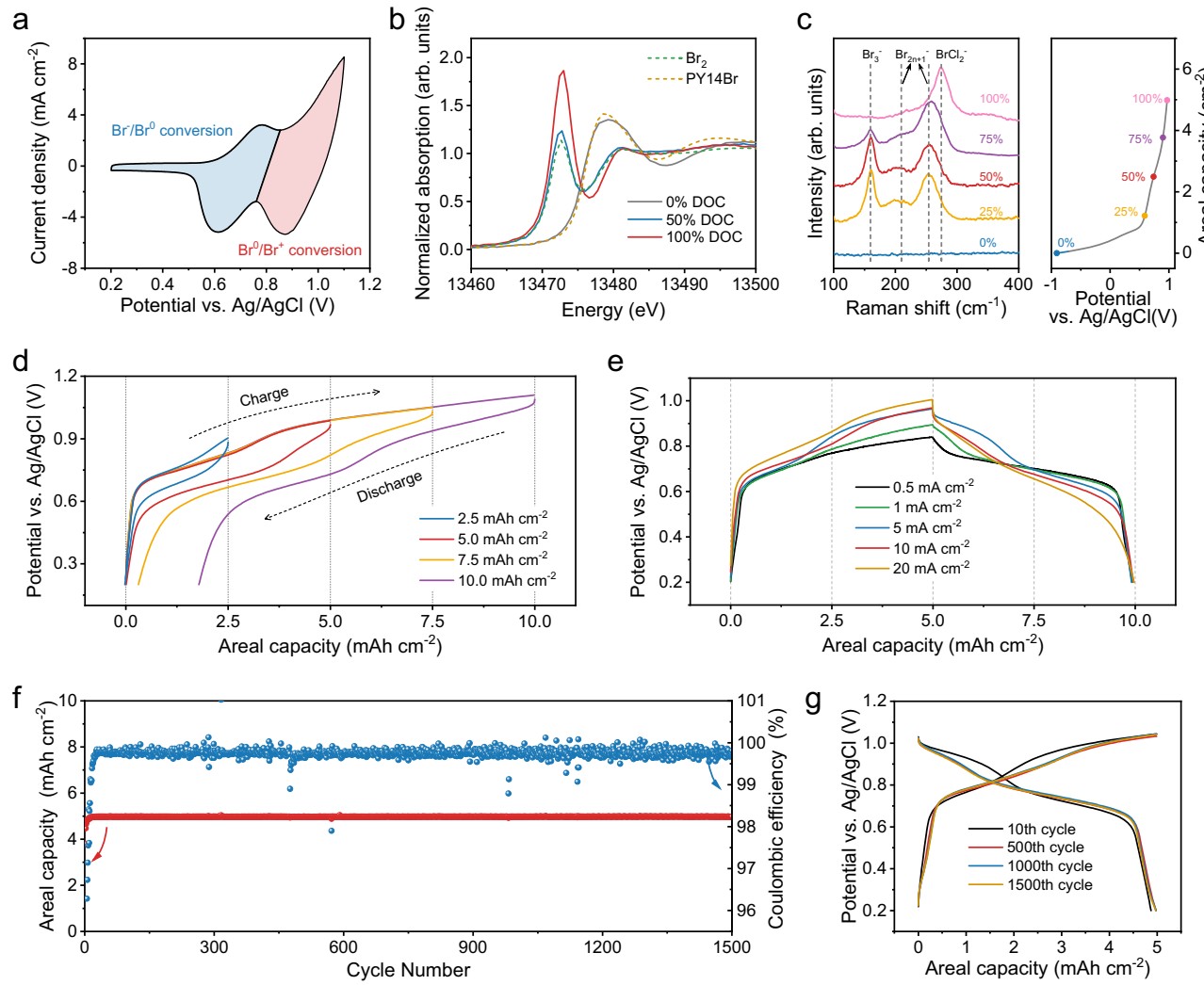

**Fig. 3 | Br⁻/Br⁰/Br⁺ conversion mechanism and performance. a** CV curve of the AC electrode in 3.25 m AlCl₃ + 1 m PY14Br at 0.1 mV s⁻¹. **b** Br K-edge XANES spectra of the AC electrode at 0% DOC, 50% DOC, and 100% DOC. Br₂-adsorbed AC and PY14Br were measured as references. **c** In situ Raman spectra of the AC electrode during a charge process. The values in percentage refer to DOC. **d** GCD profiles of the AC electrode at 5 mA cm⁻² with different capacities as the charge cutoff. **e** GCD profiles of the AC electrode at different current densities with a capacity of 5 mAh cm⁻² as the charge cutoff. **f** Cycling performance of the AC electrode in 3.25 m AlCl₃ + 1 m PY14Br at 5 mA cm⁻². **g** GCD profiles of the AC electrode at different cycles during the cycling test. Source data are provided as a Source Data file.

promote the formation of stable and water-insoluble phases, rather than facilitating their dissolution. This statement is well supported by a combination of density functional theory (DFT) calculations and GCD measurements conducted on bromide salts with various organic cations (Supplementary Fig. 23 and Supplementary Table 5).

## Br⁻/Br⁰/Br⁺ conversion performance

To reach the optimal electrochemical performance, we evaluated the GCD curves of the AC electrode at 5 mA cm⁻² with varying charge capacities (Fig. 3d). The electrode presents only one Br⁻/Br⁰ plateau at a charge capacity of 2.5 mAh cm⁻², and the high conversion reversibility can be supported by a coulombic efficiency of nearly 100%. As the reaction progresses to higher capacity, the generated polybromide species continuously modify the electrode/electrolyte interface. This modification leads to a gradual increase in potential polarization, causing the electrode potential to rise progressively throughout the Br⁻/Br⁰ conversion. At a charge capacity of 5 mAh cm⁻², reversible two-step Br⁻/Br⁰/Br⁺ conversion was evidenced by the ideal two-plateau profile with a coulombic efficiency of 99.8%. Notably, this areal capacity outperforms all the reported cathodes for AABs, such as Mn-based oxides[9,11], Prussian blue

analogs[10], polyaniline (PANI)[5], and S[10] (Supplementary Table 6). Charging electrodes to higher capacities triggered the irreversible generation of Cl₂[31], thereby resulting in severely decreased coulombic efficiency (95.9% at 7.5 mAh cm⁻² and 82.1% at 10 mAh cm⁻²).

Furthermore, an areal capacity of 5 mAh cm⁻² was employed as the charge cutoff to evaluate the effect of the charge/discharge rate on the conversion reversibility. Specifically, 0.5, 1, 5, 10, and 20 mA cm⁻² were used, referring to C-rates of 0.1, 0.2, 1, 2, and 4C, respectively (Fig. 3e). At a low current density of 0.5 mA cm⁻², charge storage of the electrode is dominated by the charge/discharge plateau referring to Br⁻/Br⁰. Upon increasing the current density, the second plateau corresponding to Br⁰/Br⁺ appears increasingly clear, albeit with a slightly elevated overpotential. The AC electrode shows a modestly increasing polarization (90 mV at 5 mA cm⁻², 127 mV at 10 mA cm⁻², and 202 mV at 20 mA cm⁻²) and IR drop (13 mV at 5 mA cm⁻², 34 mV at 10 mA cm⁻², and 57 mV at 20 mA cm⁻²) along with the current density increase, implying its fast conversion kinetics and viability for high-power devices. The fast kinetics is also verified by electrochemical impedance spectroscopy (EIS) measurements (Supplementary Fig. 24). Furthermore, the coulombic efficiency exceeds 99% at all the current densities, underscoring its superior reversibility. We also investigated the

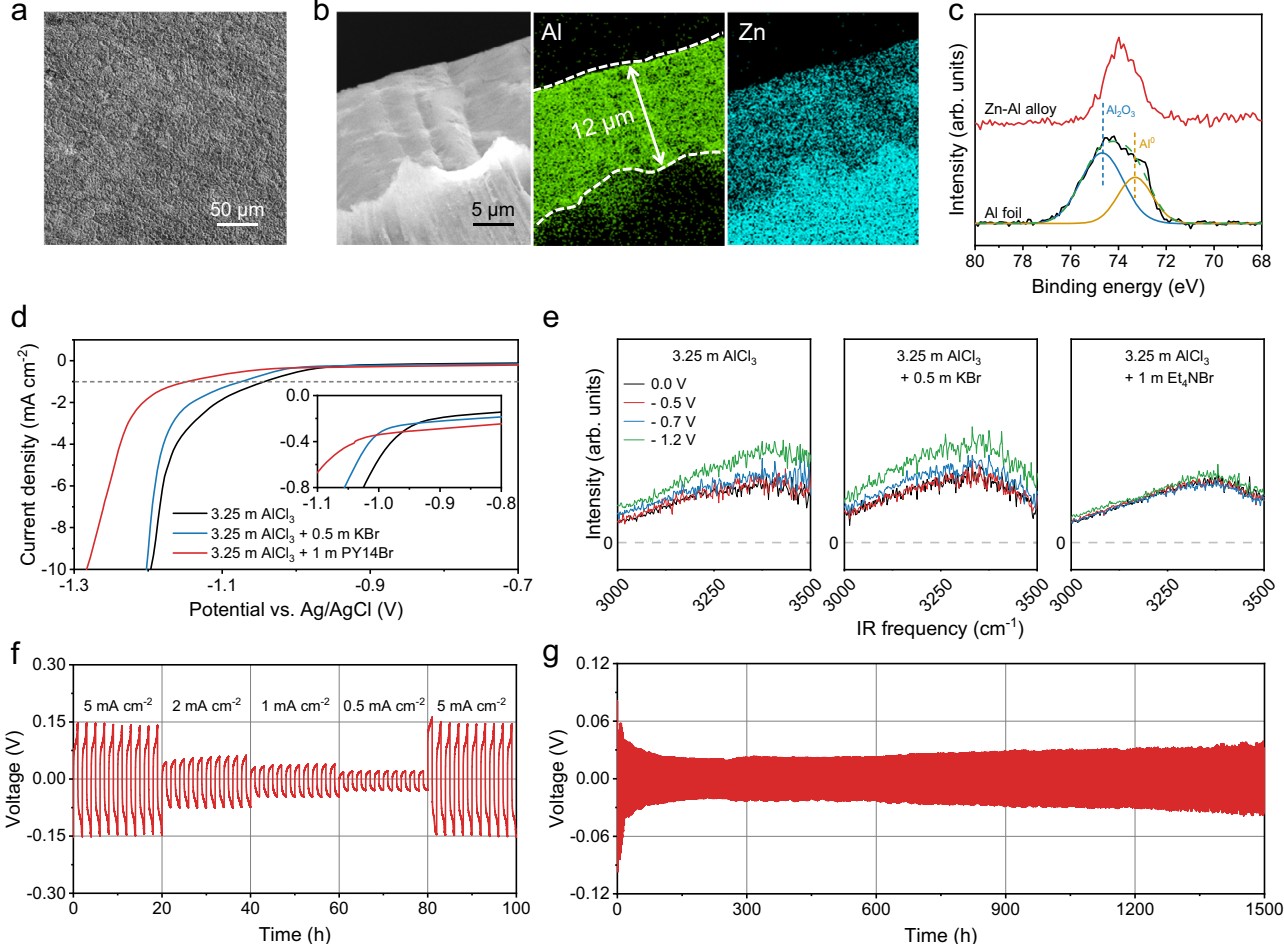

**Fig. 4 | Anode performance in hydrate-melt electrolyte.** SEM images of the Zn-Al alloy showing **a** its surface and **b** cross section, as well as the corresponding EDX mapping images. **c** Al 2p XPS spectra of Al foil and Zn-Al alloy. **d** LSV curves of the Ti electrode in 3.25 m AlCl$_3$, 3.25 m AlCl$_3$ + 0.5 m KBr, and 3.25 m AlCl$_3$ + 1 m PY14Br. **e** SFG spectra at O−H stretching frequencies of 3.25 m AlCl$_3$, 3.25 m AlCl$_3$ + 0.5 m KBr and 3.25 m AlCl$_3$ + 1 m Et$_4$NBr. **f** Galvanostatic stripping/plating of the Zn-Al||Zn-Al symmetric cell in 3.25 m AlCl$_3$ + 1 m PY14Br at different current densities. **g** Galvanostatic stripping/plating of the Zn-Al||Zn-Al symmetric cell in 3.25 m AlCl$_3$ + 1 m PY14Br at 1 mA cm$^{-2}$ and 1 mAh cm$^{-2}$. Source data are provided as a Source Data file.

rate performance by charging at 5 mA cm$^{-2}$ and discharging at different current densities. The two plateaus are observed at all discharge current densities with good conversion reversibility (Supplementary Fig. 25).

In addition, the long-term durability of the Br$^-$/Br$^0$/Br$^+$ conversion was assessed with a charge capacity of 5 mAh cm$^{-2}$. After 1500 cycles at 1 C (Fig. 3f), the discharge capacity could maintain 4.97 mAh cm$^{-2}$ with a coulombic efficiency of 99.5%. Figure 3g illustrates the GCD curves of the electrode at the 10[th], 500[th], 1000[th], and 1500[th] cycle. The AC electrode initially shows a slight decrease in polarization and subsequently maintains a constant polarization during cycling (from 115 mV at the 10[th] cycle to 90 mV at other cycles), reflecting the superior stability of the Br$^-$/Br$^0$/Br$^+$ conversion chemistry. The initial decrease in polarization can be attributed to the activation process of the electrode, which improves the electrolyte wettability of the electrode. Moreover, the electrode also exhibited high reversibility at 0.2 C, demonstrating stable operation for 350 cycles (Supplementary Fig. 26).

## Anode performance in hydrate-melt electrolytes

With the aim of assembling battery devices, it is essential to find reversible anode chemistry that can be coupled with the Br$^-$/Br$^0$/Br$^+$ conversion cathode in 3.25 m AlCl$_3$ + 1 m PY14Br. We took Zn-Al alloy as the anode, inspired by the finding that Zn-Al alloying/de-alloying exhibited considerably better kinetics, reversibility, and durability compared with direct Al stripping/plating in a 2 M aluminum triflate aqueous electrolyte[7]. In this study, the Zn-Al alloy electrode was prepared by plating Al$^{3+}$ on a Zn foil in 3.25 m AlCl$_3$ + 1 m PY14Br for 5 mAh cm$^{-2}$ at 1 mA cm$^{-2}$. Scanning electron microscope (SEM) image reveals a smooth surface of the prepared Zn-Al alloy (Fig. 4a), comparable with the original Zn foil (Supplementary Fig. 27a). This smooth surface indicates the inhibited surface corrosion during Zn-Al alloying, which is in striking contrast with the rough surface of Zn-Al alloy prepared in 3.25 m AlCl$_3$ (Supplementary Fig. 27b). Meanwhile, the cross-section energy-dispersive X-ray (EDX) spectroscopy mapping image indicates a thickness of 12 μm for the Zn-Al alloy layer (Fig. 4b). Based on the quantification result from inductively coupled plasma optical emission spectrometry (ICP-OES), the Zn/Al atomic ratio of the Zn-Al alloy layer is about 1:6 (Supplementary Table 7).

The chemical composition of the Zn-Al alloy layer was analyzed using X-ray photoelectron spectroscopy (XPS). Figure 4c compares the Al 1s XPS spectra of the Zn-Al layer and Al foil. The pronounced surface passivation of the Al foil by an oxidation layer is evidenced by the characteristic Al$_2$O$_3$ peak at 74.7 eV. Moreover, the Al 1s XPS peak of Zn-Al alloy is located at a notably higher binding energy (74.0 eV) than Al$^0$ in the Al foil (73.4 eV)[8]. This observation reflects that Al in Zn-Al alloy presents electron deficiency compared with Al foil, suggesting a higher Zn-Al alloying potential than direct Al plating. This higher

potential could play a crucial role in inhibiting side reactions like hydrogen evolution and metal corrosion.

To evaluate the electrochemical behavior, we first employed a three-electrode Swagelok cell with an over-capacity AC electrode as the counter electrode and an Ag/AgCl reference electrode. By using a Ti foil as the working electrode, the cathodic stability of 3.25 m AlCl$_3$, 3.25 m AlCl$_3$ + 0.5 m KBr, and 3.25 m AlCl$_3$ + 1 m PY14Br was assessed using an LSV test at 10 mV s$^{-1}$. Throughout the test, we confirmed the absence of Al deposition (Supplementary Fig. 28), likely attributed to the large potential gap between Al deposition and hydrogen evolution on Ti. Evidently, 3.25 m AlCl$_3$ + 1 m PY14Br demonstrates the best cathodic stability, showing significantly suppressed side reactions like hydrogen evolution (Fig. 4d). For instance, using 1 mA cm$^{-2}$ as the benchmark, the cathodic stability of 3.25 m AlCl$_3$ + 1 m PY14Br reaches −1.14 V vs. Ag/AgCl, considerably lower than those of 3.25 m AlCl$_3$ (−1.04 V vs. Ag/AgCl) and 3.25 m AlCl$_3$ + 0.5 m KBr (−1.07 V vs. Ag/AgCl). Another hydrate-melt electrolyte, 3.25 m AlCl$_3$ + 1 m Et$_4$NBr, which has similar structure with 3.25 m AlCl$_3$ + 1 m PY14Br and also allows for reversible Br$^-$/Br$^0$/Br$^+$ conversion, presented a similar behavior at the anode side (Supplementary Fig. 29).

To confirm the suppression of hydrogen evolution in hydrate-melt electrolytes, we performed sum frequency generation (SFG) vibrational spectroscopy. SFG spectroscopy is surface-specific due to its selection rules, enabling selective probing of the electrode/electrolyte interface without interference from the bulk electrolyte due to the SFG selection rule[32–34]. We focused on the SFG signal of interfacial water (O-H stretching at 3000−3550 cm$^{-1}$), whose intensity is a direct indicator of the occurrence of hydrogen evolution at the electrode/electrolyte interface with a current density below 1 μA cm$^{-2}$ [32,35]. An optically transparent electrode (SiO$_2$-supported graphene electrode) was used as the working electrode to ensure optical accessibility for SFG. We investigated 3.25 m AlCl$_3$ + 1 m Et$_4$NBr, comparing it with 3.25 m AlCl$_3$ and 3.25 m AlCl$_3$ + 0.5 m KBr (Fig. 4e). The SFG signal starts to increase at around −0.7 V vs. Ag/AgCl in 3.25 m AlCl$_3$ and 3.25 m AlCl$_3$ + 0.5 m KBr, indicating the occurrence of hydrogen evolution. In contrast, the SFG signal change is negligible in 3.25 m AlCl$_3$ + 1 m Et$_4$NBr even at a potential of −1.2 V vs. Ag/AgCl, confirming substantially suppressed hydrogen evolution. In addition, the intensity of the SFG signal in 3.25 m AlCl$_3$ + 1 m PY14Br also remains stable along the potential decrease, indicating consistent trend of hydrogen evolution suppression despite the apparent noise (Supplementary Fig. 30).

To assess the Zn-Al alloying/de-alloying kinetics and durability, we employed a two-electrode Swagelok cell with a symmetric Zn-Al||Zn-Al configuration. After an interfacial activation step at 5 mAh cm$^{-2}$ for 100 h (Supplementary Fig. 31), we collected voltage profiles of the cell during Zn-Al alloying/de-alloying at a variety of current densities in 3.25 m AlCl$_3$ + 1 m PY14Br (Fig. 4f). The cell demonstrated nearly identical overpotential between the initial and later profiles at 5 mAh cm$^{-2}$, indicating its high reversibility. The hysteresis voltage of the cell ranges from 69 ~ 150 mV at 5 mA cm$^{-2}$, evidencing fast Zn-Al alloying/de-alloying kinetics. Moreover, the Zn-Al||Zn-Al cell achieved a cycle life of more than 1500 h at 1 mA cm$^{-2}$ and 1 mAh cm$^{-2}$ with a hysteresis voltage of less than 40 mV (Fig. 4g), in contrast to the rapid failure of the Al||Al cell in 3.25 m AlCl$_3$ + 1 m PY14Br (Supplementary Fig. 32). Even at a harsh operating condition of 5 mA cm$^{-2}$ and 5 mAh cm$^{-2}$, the Zn-Al||Zn-Al cell can stably operate for 500 h with an average hysteresis voltage of 100 mV (Supplementary Fig. 33). The presence of soft short-circuiting was excluded through temperature-dependent galvanostatic stripping/plating tests of a Zn-Al||Zn-Al symmetric cell using 3.25 m AlCl$_3$ + 1 m PY14Br (Supplementary Fig. 34).

We further conducted a metal stripping experiment on the Zn-Al alloy electrode in 3.25 m AlCl$_3$ + 1 m PY14Br at 5 mA cm$^{-2}$ for 1 h to evaluate Zn participant. A three-electrode setup was used, consisting of a Zn-Al alloy working electrode, an over-capacity AC electrode as

the counter electrode, and an Ag/AgCl reference electrode. The electrolyte was then extracted and analyzed via ICP-OES to quantify the stripped Zn. The results indicated minimal Zn participation, accounting for only 3.9% of the total Zn-Al alloy capacity. In addition, we found that the Zn-Al||Zn-Al cell could also function in 3.25 m AlCl$_3$ (Supplementary Fig. 35) and 3.25 m AlCl$_3$ + 0.5 m KBr (Supplementary Fig. 36), but with a short cycling life of less than 100 h. SEM images reveal the heavy surface corrosion of Zn-Al alloy after only ten cycles in 3.25 m AlCl$_3$ and 3.25 m AlCl$_3$ + 0.5 m KBr, while Zn-Al alloy maintains a smooth surface in 3.25 m AlCl$_3$ + 1 m PY14Br (Supplementary Fig. 37), highlighting the anti-corrosion capability of 3.25 m AlCl$_3$ + 1 m PY14Br.

### Demonstration of aqueous Al-Br cells

Given the desirable compatibility of the 3.25 m AlCl$_3$ + 1 m PY14Br electrolyte with both Zn-Al alloy anode chemistry and Br$^-$/Br$^0$/Br$^+$ conversion cathode chemistry, we assembled aqueous Al-Br cells using two-electrode Swagelok cells. A Zn-Al alloy anode (20 μm in thickness) and an AC hosting cathode (278 μm in thickness), both capable of achieving an areal capacity of 5 mAh cm$^{-2}$, were coupled with the 3.25 m AlCl$_3$ + 1 m PY14Br electrolyte. In the cell, the stabilization effect of bulky organic PY14$^+$ cations could reduce polyhalide dissolution into the electrolyte and minimize their shuttling to the anode, thus effectively mitigating Zn-Al alloy anode corrosion by oxidative polyhalide species (Supplementary Fig. 38) The high electrochemical stability of PY14$^+$ cations was confirmed during cell operation (Supplementary Fig. 39). Following the three-electrode measurement for the Br$^-$/Br$^0$/Br$^+$ conversion, we employed 5 mAh cm$^{-2}$ as the charge cutoff. After 10 charge/discharge cycles, negligible Zn was detected in the electrolyte by ICP-OES indicating that Zn has rare impact on the electrochemical behavior of Br in the battery. The Br utilization efficiency for the Al-Br cell was evaluated by controlling the electrolyte amount (Supplementary Fig. 40). With Br utilization efficiencies ranging from 10% to 40%, the Al-Br cell exhibits nearly identical GCD curves with two distinct charge/discharge plateaus. In our following test, a Br utilization efficiency of 10% was fixed.

Figure 5a presents the GCD curve of the device at a large current density of 20 mA cm$^{-2}$, which exhibits a high energy efficiency of 78% and a small mid-capacity voltage polarization of 300 mV. In addition, the device demonstrated reversible operation at varying rates ranging from 5 to 20 mA cm$^{-2}$, corresponding to a C-rate range of 1 - 4 C (Fig. 5b, Supplementary Figs. 41, 42). Compared with the three-electrode tests, the Al-Br cell shows slightly higher polarization at the same current densities (135 mV at 5 mA cm$^{-2}$, 176 mV at 10 mA cm$^{-2}$, and 300 mV at 20 mA cm$^{-2}$). This polarization enlargement is attributed to the additional overpotential from the Zn-Al alloy anode. Based on the GCD measurements, our Al-Br cell achieved an average discharge voltage of 1.7 V. With an areal capacity of 4.89 mAh cm$^{-2}$, the device delivered an areal energy density of 8.31 mWh cm$^{-2}$, at least twice some of recently relevant reports for aqueous Zn batteries (AZBs) (0.06 ~ 3.61 mWh cm$^{-2}$), AABs (0.08 ~ 0.99 mWh cm$^{-2}$), and non-aqueous Al batteries (NABs) (0.18 ~ 2.05 mWh cm$^{-2}$) (Fig. 5c and Supplementary Table 8).

We further estimated the volumetric energy and power densities of our Al-Br cell based on the total volume of the two electrodes and the separator (Fig. 5d and Supplementary Fig. 43). Impressively, our cell exhibits an energy density of 267 Wh L$^{-1}$, which is well comparable to commercial Li-ion batteries (263 Wh L$^{-1}$, Panasonic CG420A), and considerably higher than lead-acid batteries (110 Wh L$^{-1}$)[36] and Li thin-film micro-battery (2.8 Wh L$^{-1}$)[37]. Besides, the maximum power density of our Al-Br cell reaches 1069 W L$^{-1}$, within the range of commercial electrochemical capacitors (93 ~ 1409 W L$^{-1}$)[38]. These enhanced energy-power merits can be attributed to the capacity-dense Zn-Al alloy anode and high-loading cathodes, both exhibiting rapid electrochemical kinetics.

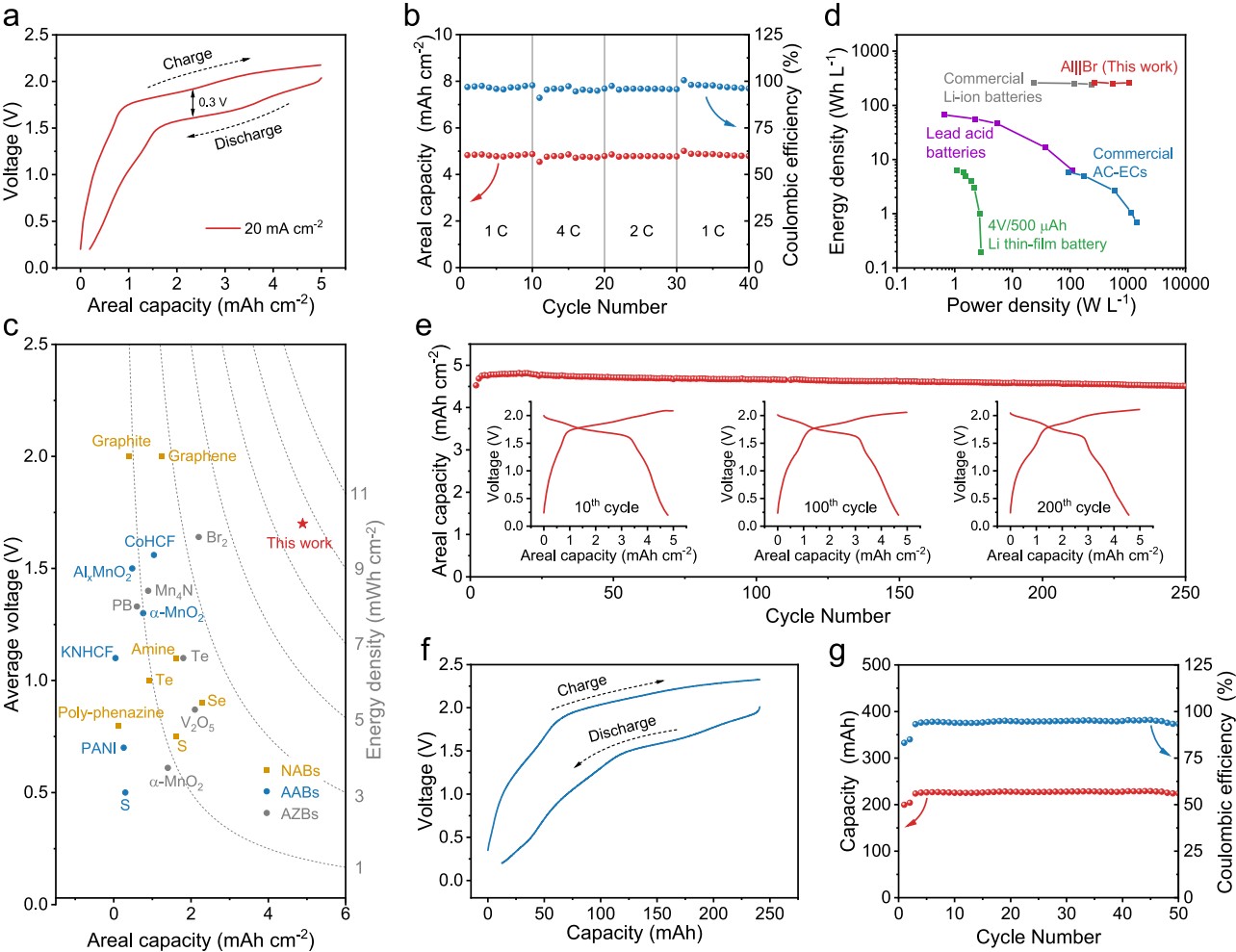

**Fig. 5 | Electrochemical performance of aqueous Al-Br cells. a** GCD curve of the cell at 20 mA cm⁻². **b** Rate performance of the cell at varying current densities. **c** Areal capacity of the cell as a function of the average discharge voltage. The Al-Br cell is compared with recently reported AZBs (gray dots), AABs (blue dots), and NABs (yellow dots). The source of the literature data shown in this figure can be found in Supplementary Table 8. **d** Ragone plots of our Al-Br device in comparison with state-of-the-art electrochemical energy storage devices. **e** Cycling performance of the cell at 5 mA cm⁻². The inset displays GCD profiles of the cell at the 10th, 100th, and 200th cycle. **f** GCD curve of the pouch-cell Al-Br device at 5 mA cm⁻². **g** Cycling performance of the pouch-cell Al-Br device at 5 mA cm⁻². Source data are provided as a Source Data file.

The long-term operational durability of the Al-Br cell was assessed at 5 mA cm⁻² (Fig. 5e). The polarization initially decreases and then increases with cycling (161 mV at the 10th cycle, 149 mV at the 100th cycle, and 170 mV at the 200th cycle). The increasing polarization is likely due to the enlarged electrode resistance as cycling progresses. After 250 cycles, the device maintained stable operation with a GCD profile shape nearly identical along the cycling test. Additionally, we identified the good temperature tolerance of the Al-Br cell within 298.15 - 333.15 K (Supplementary Fig. 44), which is attributed to the structural stability of the electrolyte at elevated temperatures (Supplementary Fig. 45).

Finally, we fabricated pouch-cell Al-Br devices (4 × 6 cm²) as the proof of concept, which comprised a two-layer AC cathode and a one-layer Zn-Al alloy anode (Supplementary Figs. 46–47). As shown in Fig. 5f, the device was able to achieve a capacity of 228 mAh at a current density of 5 mA cm⁻². Additionally, it demonstrated good rechargeability over 50 cycles and successfully powered a timer with an operating voltage of 1.5 V (Fig. 5g and Supplementary Fig. 48).

## Discussion

In summary, our study has showcased the design of hydrate-melt electrolytes consisting of cost-effective AlCl₃ and organic halide salts such as PY14Br, which are featured by strongly ion-associated water molecules and loosely bound halogen anions. Experimental and theoretical studies unveil that the vast majority of water molecules (>98%) in the electrolyte engaged in strong ion solvation, leading to significantly reduced water activity and an enlarged electrochemical stable potential window. Importantly, the electrolyte exhibited good compatibility with both Br⁻/Br⁰/Br⁺ conversion, offering large areal capacity (5 mAh cm⁻²), and the Zn-Al alloying/de-alloying chemistry, displaying minimal polarization (~100 mV at 5 mA cm⁻²) and maintaining a smooth electrode surface. Combining these cathode and anode chemistries with the hydrate-melt electrolyte in a single cell enabled us to demonstrate an aqueous Al-Br battery device, which achieved a high average discharge voltage of 1.7 V. Leveraging the large areal capacity and rapid electrochemical kinetics of both the anode and cathode, the cell delivered a volumetric energy density comparable to commercial Li-ion batteries (267 Wh L⁻¹), while exhibiting a large volumetric power density (1069 W L⁻¹) approaching electrochemical capacitors. These results highlight the potential of the proposed hydrate-melt electrolyte as a crucial component in advancing the development of AABs. Meanwhile, it is essential to acknowledge the persistent challenges encountered in the constructed Al-Br cell, including side reactions such as hydrogen evolution, Zn dissolution at

the anode, and Br-species dissolution at the cathode, as well as potential issues related to anode-cathode cross-talking. We hope that our encouraging results will inspire further efforts aimed at addressing these remaining challenges and promoting the practicality of such a battery system through continued advancement of the interphase and electrolyte.

## Methods

### Chemicals

Aluminum chloride hexahydrate ($AlCl_3 \cdot 6H_2O$, 99%), potassium chloride (KCl, ≥99.8%), potassium bromide (KBr, ≥99.0%), tetraethylammonium chloride ($Et_4NCl$, ≥98%), tetraethylammonium bromide ($Et_4NBr$, ≥98%), 1-butyl-1-methylpyrrolidinium chloride (PY14Cl, ≥99.0%), 1-butyl-1-methylpyrrolidinium bromide (PY14Br, ≥99.0%), polytetrafluorethylen (PTFE, 60 wt % dispersion in $H_2O$), glass fiber membrane (0.02 mm thickness, 0.7 μm pore size), and super P, N-methyl-2-pyrrolidone (≥99.8%) were purchased from Sigma Aldrich. Zn foil (20 μm in thickness) and Al foil (25 μm in thickness) were purchased from Thermo Fisher Scientific. Graphene oxide (GO) was purchased from GaoxiTech Co., Ltd. Carbon cloth was purchased from The Fuel Cell Store. AC was purchased from Kuraray Co., Ltd (YP-80F) which possesses a microporous structure with specific surface area of $2271 \, m^2 \, g^{-1}$ and bulk density of $0.18 \, g \, ml^{-1}$. All chemicals were directly used without further purification.

### Preparation of hydrate-melt electrolytes

The hydrate-melt electrolytes were prepared by dissolving $AlCl_3 \cdot 6H_2O$ and organic halide salts in deionized water. The concentration of $AlCl_3$ was controlled at 3.25 m and the concentration of organic halide salts was controlled on demand. The solution was heated at 80 °C for 1 h to facilitate dissolution. After preparation, the electrolytes were stored and evaluated exclusively at room temperature.

### Preparation of Zn-Al alloy anode

An asymmetric cell was assembled by using Zn foil as the working electrode, Al foil as counter electrode, and 3.25 m $AlCl_3$ + 1 m PY14Br as electrolyte to prepare Zn-Al alloy anode. The asymmetric cell was charged at $1 \, mA \, cm^{-2}$ for 5 h. During this electrochemical process, $Al^{3+}$ from the electrolyte reacted with the Zn foil to form Zn-Al alloy.

### Electrochemical measurements

To prepare the AC electrode, YP80-F and PTFE with a mass ratio of 9:1 were dispersed in water with a mass ratio of 9:1, milled homogeneously, and dried in vacuum oven at 80 °C for 12 h. The volume of water we added was around $50 \, mL \, g_{PTFE}^{-1}$. The mortar and pestle were made from agate. Afterwards, a self-standing film with a mass loading of $15 \, mg \, cm^{-2}$ and a thickness of 278 μm was obtained. To measure the electrolyte potential window, three-electrode Swagelok cells were adopted with Ti foil (12 mm in diameter) as the working electrode, over-capacity AC (8 mm in diameter, ~1 mm in thickness) as the counter electrode, and Ag/AgCl reference electrode. Two Ti rods (99.5% purity with base area of $0.79 \, cm^{-2}$) were used as the current collector for the working electrode and counter electrode, respectively. In the evaluation of $Br^-/Br^0/Br^+$ conversion, the AC electrode with a mass loading of $15 \, mg \, cm^{-2}$ (278 μm in thickness, 7 mm in diameter) was employed as the working electrode, while over-capacity AC electrode (8 mm in diameter, ~1 mm in thickness) was used as the counter electrode. Two layers of glass fiber membrane were used as separators. To evaluate the Zn-Al alloy electrode (8 mm in diameter), the same over-capacity AC electrode was used as the counter electrode.

In the Al-Br cell test, the AC electrode with a loading mass of $15 \, mg \, cm^{-2}$ (278 μm in thickness, 7 mm in diameter) and the Zn-Al alloy (20 μm in thickness, 8 mm in diameter) were used as the cathode and anode, respectively. A Ti rod and a stainless-steel rod (SS304 with base area of $0.79 \, cm^{-2}$) were used as the current collectors for the cathode and anode, respectively. A spring (SS304 with force constant of $4.2 \, N \, mm^{-1}$) was used in Swagelok cell to fix the electrode on the current collectors. GO-coated glass fiber membrane was used as the separator. Specifically, GO was mixed with deionized water to obtain GO aqueous solution with a concentration of $0.5 \, mg \, ml^{-1}$. Then the membrane was immersed in the solution for five minutes and dried under vacuum oven overnight. The devices were assembled in two-electrode Swagelok cells. Pouch-cell Al-Br devices with an open hole vent were assembled by using a two-layer AC cathode ($4 \times 6 \, cm^2$), a one-layer Zn-Al alloy anode ($4 \times 6 \, cm^2$), and GO-coated glass fiber separator (20 μm). Ti foil was used as the cathode current collector.

CV, LSV, and EIS results were collected by a VMP3 Multichannel Potentiostat, Biologic. All electrochemical measurements were conducted at room temperature (298.15 K). The EIS measurements were carried out at a 20 mV AC oscillation amplitude over the frequency range of 100 kHz to 0.01 Hz. GCD measurements were performed with a Land battery test system (LAND CT2001A). The energy density ($E$) is calculated based on the GCD profiles by Eq. (2), where $I$ is the applied current density, $U$ is the cell output voltage, $t$ is the discharging time and $V$ is the volume of anode, cathode and separator.

$$E = I \int_0^t U(t) dt / V \qquad (2)$$

The ionic conductivity (σ) of the electrolytes was measured by EIS at 298.15 K. Specifically, a sealed 2-electrode cell consisting of two platinum electrodes placed parallel to one another with a nominal constant (C) of $1.021 \, cm^{-1}$ was used. The EIS measurements were carried out at a 20 mV AC oscillation amplitude over the frequency range of 100 kHz to 0.01 Hz. Ionic conductivities were calculated according to Eq. (3), where R represents the resistance of the electrolyte.

$$\sigma = \frac{C}{R} \qquad (3)$$

### Characterization

Scanning electron microscope (SEM), energy-dispersive X-ray spectroscopy (EDX), and elemental mapping images were recorded on a field-emission scanning electron microscope Zeiss Gemini 500 equipped with an Oxford XmaxN-150 EDX detector. UV-vis spectra were obtained on Agilent Cary 5000. X-ray photoelectron spectroscopy (XPS) was studied by a multiprobe system (Scienta Omicron) coupled with an Al Kα source and an electron analyzed (Argus CU) with 0.6 eV resolution. (In situ) Raman spectra were obtained from WITec alpha300 R. Electrolyte viscosities were measured by rotational viscometer. Inductively coupled plasma optical emission spectrometry (ICP-OES) was conducted by Avio 220 Max. Fourier-transform infrared (FT-IR) spectra were obtained from Bruker Tensor II spectrometer. The electrochemical sum frequency generation (SFG) spectroscopy measurements were conducted using a noncolinear SFG setup[35]. The spectra were collected under *ssp* polarization combinations, where *ssp* indicates s-polarized SFG, s-polarized visible, and p-polarized IR beams, and are normalized against the SFG spectrum of the $SiO_2$/gold. A height displacement sensor (CL-3000, Keyence) was used to check the sample height upon flowing electrolyte solutions. Each spectrum was acquired with an exposure time of 600 s and measured more than three times on average to minimize the system error. The electrochemical SFG spectroscopy was performed in a three-electrode system. Ag/AgCl was used as the reference electrode and a gold wire as the counter electrode. The working electrode was a $SiO_2$-supported graphene electrode. The potentials were applied by an electrochemical workstation (Metrohm Autolab PGSTAT302) during the electrochemical SFG measurements.

Dielectric relaxation and loss spectra at $0.8 \leq v/GHz \leq 36$ were measured using a frequency domain reflectometer based on an Anritsu Vector Star MS4647A. Spectra at $56 \leq v/GHz \leq 125$ were recorded analogously using an external Anritsu 3744 A mmW module[39,40]. Calibration of the reflectometry system was carried out using air, water, and conductive silver paint.

XAS measurements were conducted at beamline P65 (DESY, Hamburg) and BL22 (ALBA, Barcelona). The measurements were performed at the K edge of Br (13.4–13.6 keV). The data were acquired in transmission mode under room temperature. The AC electrode was charged to the specific potentials after 10-cycle activation. Then the cathodes were taken out and encapsulated in Kapton for XAS test. XAS data were processed and analyzed using the Demeter and Athena software package[41,42].

WAXS measurements were conducted at beamline P62 (DESY, Hamburg) and BL11 (ALBA, Barcelona). The electrolytes were added into borosilicate capillaries and sealed with paraffin wax in a glovebox. Subsequently, the capillaries were mounted on a sample holder without temperature control. The measurements were performed at a $q$ range of 0.6 - 3 Å⁻¹ for WAXS. All samples were measured for 30 s and 5 times to exclude beam damage.

## Theoretical simulations

All ab initio molecular dynamics (AIMD) simulations were conducted using the Vienna Ab Initio Simulation Package (VASP) within the framework of density functional theory (DFT) calculations[43]. The Perdew-Burke-Ernzerhof (PBE) exchange-correlation functional was employed for these calculations[44]. The first Brillion zone k-point sampling employed the gamma-centered scheme with the $1 \times 1 \times 1$ k-meshes. The computational supercell in each case comprised ~850 atoms, with densities matching with the experimentally measured values. Periodic boundary conditions were applied in all directions to simulate bulk electrolytes. Initial configurations of the electrolytes were generated by randomly placing ions and molecules within the supercell with arbitrarily chosen orientations. Long-range van der Waals dispersion interactions were employed using the DFT-D2 method[45]. Each electrolyte system was equilibrated at 298.15 K in the canonical ensemble (NVT) for 5 ps. Constant temperature conditions were maintained using a Nose-Hoover thermostat. A time step of 0.5 fs was used.

Radial distribution function (RDF) $g(r)_{A-B}$ was evaluated using Eq. (4), where $\rho_b$ represents the average density of type-B atoms within spherical shells of type-A atoms. The coordination number was obtained by integrating the RDF of each solvation shell. The value of coordination number, N(r), was computed using Eq. (5), where $\rho_N$ denotes the average number density of the surrounding atoms[22].

$$g(r)_{A-B} = \frac{1}{\rho_b} \frac{1}{N_A} \sum_{i=1}^{N_A} \sum_{J=1}^{N_B} \frac{\delta(r_{ij}) - r}{4\pi r^2} \quad (4)$$

$$N(r) = 4\pi \rho_N \int_0^r r^2 g(r) dr \quad (5)$$

DFT calculations were conducted by VASP with PBE exchange-correlation functional. The first Brillion zone k-point sampling employs the gamma-centered scheme with the $1 \times 1 \times 1$ k-meshes.

## Data availability

Source data are provided with this paper.

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

## Acknowledgements

This work was financially supported by European Union's Horizon Europe research and innovation programme (LEAF 101186701), German Research Foundation (DFG) within the Cluster of Excellence, and CRC 1415 (Grant No. 417590517). X.C. was supported by a grant from the China Scholarship Council (File No. 202108060037). The authors acknowledge the use of the facilities in the Dresden Center for Nanoanalysis (DCN) at the Technische Universität Dresden, computational support through the Center for Information Services and High-Performance Computing (ZIH) at TU Dresden and the Supercomputing Center of Max Planck Computing & Data Facility (MPCDF), beam time allocation at beamline P62 and P65 at the PETRA III synchrotron of DESY (Germany) and beamline BL11 and BL22 at the ALBA Synchrotron (Spain).

## Author contributions

X.C., X.F. and M.Y. conceived the project and electrolyte design principle and experiments. X.C. and J.D. conducted material preparation, theoretical calculations, electrochemical testing, and most characterizations. Y.W., J.H., Y. H. and M.B conducted SFG spectra, dielectric relaxation spectra and dielectric loss spectra measurement. J.Z. prepared the schematics. X-D.L. supervised theoretical calculations. J.Z. and J.L. performed Raman measurements. J.D., X-H.L., D.L. conducted XAS measurements. A.M. performed WAXS measurements, Q. G. conducted XPS measurements. X.C. and M.Y. wrote the whole manuscript. All authors contributed to discussion and revising the manuscript under the supervision of X.F. and M.Y.

## Funding

## Competing interests

The authors declare no competing interests.
