## [Transparent Peer Review file · Nature Communications]

Hydrate-Melt Electrolyte Design for Aqueous Aluminium-Bromine Batteries with Enhanced Energy-Power Merits

Corresponding Author: Professor Xinliang Feng

Version 0:

Reviewer comments:

Reviewer #1

(Remarks to the Author)

The idea is somewhat interesting but not so much because there are many papers with similar approaches out there. The manuscript is also not well prepared for scientific publication due to some of the following reasons:

- 1) the scientific elaboration and arguments are lacking
- 2) many of the claims in this work are overstated
- 3) many of the claims are jumping into conclusions rather than stating data-based observation

These reasons reduce its proper readiness for scientific readership.

Moreover, the use of 'melt electrolyte' term in the title is also misleading and looks more like a clickbait. Just because the organic halide salts being used in this work (KBr, PY14Br, Et4NBr) were heated at 80°C to facilitate dissolution, which is also a relatively low temperature (< 100°C), this 'part' of the electrolyte is not proper to be called as 'melt'. They are just normal Br-based ionic liquids.

This work attempted to offer a solution to increase the energy and power performance of Al-ion batteries in general, which is by introducing another ion into the hydrate AlCl₃ electrolyte system, by adding organic bromide salts. Authors stated that by adding Br into the system, the two-step Br conversion reaction (Br⁻/Br⁰/Br⁺) may result in "superb reaction kinetics that could potentially allow for a practical areal capacity akin to metal stripping/plating reactions". Authors stated that the first Br⁻/Br⁰ reaction step may "overcomes the hindrance associated with solid-state Al³⁺ diffusion during the charge storage process". First of all, these two statements are not supported with the data presented by Authors. Furthermore, Authors also stated that "the second Br⁰/Br⁺ reaction tends to be initiated with the participation of Cl⁻ through interhalogen reactions, forming Br-Cl compounds" – which is also a statement not investigated nor supported with Authors' data. There is no analytical data that presents the existence of any Br-Cl compound. To construct the full cell, Authors also introduced Zn-Al alloy as the anode, which practically means that there is a possibility that Zn may also participate in the reaction – also something that Authors exclude in their investigation.

In its current form, I think the investigation presented in this work lacks the necessary comprehensiveness and systematic approach. As such, I cannot recommend it for publication, particularly for the audience of Nature Communications. To address these issues, the authors should conduct a more thorough and in-depth study, providing clear explanations, detailed analyses, and further data-based scientific elaboration. Additionally, revising the writing to ensure it is more objective and data-driven is essential to meet the standards of scientific publication.

Specific comments:

1. Page 6, line 126: Authors stated that "(Fig. 2b), ... indicates the compatibility of 3.25 m AlCl₃ with Br⁰/Br⁺ conversion chemistry. This is one example on how Authors are overstating their results and making claims not related to their data. The measurement setup in this case are very simple and standard (WE: Ti rod, RE: Ag/AgCl, CE: AC) with 3.25 m AlCl₃ as the electrolyte – no Br in the system. How can this prove this system's compatibility with Br while it is not there?
2. Page 7, line 172: Authors stated that the addition of 1 m PY14Br breaks the hydrogen bonding network among water, contributing to suppressing water activity. Then how can double sheath of solvation layer majorly filled with water molecules (as Authors also presented in Figure 1) can be formed?
3. Page 8, line 173: Authors stated 'A similar conclusion can also be drawn (Supplementary Fig. 8)', but did not elaborate further. Please at least make a summarizing sentence in the main manuscript or elaborate clearly in supporting information

to make the data presentation meaningful.

4. Similarly, please elaborate further on Supplementary Figure 9 as well, which contains three sub-figures (9a-9c). What do Authors want to say or present by providing these snapshots? At least make some annotations or marks on what are the important parts of those images.

5. Similarly, please elaborate on Supplementary Figure 10, which contains six sub-figures (10a-10f). There must be some important points of these images so that the experiment was carried out that it has to be presented. Please use the 'Supplementary Information' field wisely and responsibly.

6. Similarly, please elaborate further on Supplementary Figure 11, which contains three sub-figures (11a-11c). What do Authors want to say or present by providing these snapshots? At least make some annotations or marks on what are the important parts of those images. One suggestion is by marking which area is the inner or outer solvation sheath.

7. Page 8 line 184: Authors stated that '... dominant Br⁻ (75%) and Cl⁻ (79.5%) anions are revealed to reside in the outer solvation sheath of Al³⁺', but where is Br in Figure 2g? If adding Br was missed in the modeling, I think Authors need to repeat everything by including Br in all cases, not just by assuming Cl⁻ and Br⁻ as the same thing just because both are halide ions. Possible differences in terms of charge density, polarizability, ionic size, solubility, oxidizing power, etc, may affect the system. Even if Authors think that it will end up not having any difference, their similarity should be proven with data.

8. Moreover, did Authors also include 'temperature' in the modeling process? Authors stated that the organic halide salts being used in this work (KBr, PY14Br, Et4NBr) were heated at 80°C to facilitate dissolution in AlCl₃. Was the elevated temperature just used for dissolution purposes? Are the electrolyte mixtures used at 80°C during the whole electrochemical experiments or are they cooled down first before being used? If they are cooled down, how long after they are being cooled down do precipitates occur? Also, if they are considered at high concentrations, what about their viscosity?

9. Page 8, line 190: Authors stated: 'These loosely bound Br⁻ and Cl⁻ anions are expected to facilitate fast interfacial charge transfer during Br⁻/Br₀/Br⁺ conversion by avoiding kinetics-limiting anion-cation dissociation.' Is there any data that can support this claim?

10. Page 8, line 192: Authors stated: "...the statistical analysis of water molecules involved in strong solvation (including the two solvation sheaths of Al³⁺ and one solvation sheath of other ions) reveals minimal free water in 3.25 m AlCl₃ (6.1%), 3.25 m AlCl₃ + 0.5 m KBr (5.4%), and 3.25 m AlCl₃ + 1 m PY14Br (1.8%), underscoring the hydrate-melt nature of all three electrolytes.". How did Authors derive the statistical analysis and where is the data? A detailed description of how these are calculated and derived should be disclosed and discussed as well.

11. Page 8, line 200: Authors stated that there are two pairs of redox peaks on the CV, but on the data, there are just three peaks (Figure 3a). One pair (two peaks) in the 0.6 – 0.7 region, which Authors assigned to Br⁻/Br₀ conversion, while there is only one single peak between 0.8 – 1.0 V (vs. Ag/AgCl). Isn't this evidence of irreversibility of the Br₀/Br⁺ conversion? Moreover, isn't the sudden steep current increment past 1.0 V indicative of OER?

12. Page 8, line 202: Authors stated: "These two potential regions are consistent with the potentials of Br⁻/Br₀ and Br₀/Br⁺ conversion". Please elaborate on this statement specifically and clearly, as to where each conversion reaction step occurs (similar to what Authors did on page 9 line 211-213). This is important now because the provided CV data does not support this claim well.

13. For Figure 3b, why are the peaks at 13470 eV not 'smooth' and looked more like a triangle (as if some data are missing, not high-resolution data), not like the peak beside it at around 13480 eV which is smooth?

14. For Figure 3c, I suggest Authors not only give dashed lines but also give annotations to provide better instructiveness of the data to the reader. What is the point of this image?

15. In Figure 3e, it seems like starting from 5 mA cm⁻² to 20 mA cm⁻², there are significant iR drops during the charge-discharge test. How do Authors comment on this?

16. At 1 C, the stability of the AC electrode reached 1500 cycles (Figure 3f), then why at 0.2 C the stability dropped to only 200 cycles (Supplementary Figure 15)?

17. Page 10, statement in Line 230-235 (In 3.25 m AlCl₃ + 1 m PY14Br, the PY14⁺ cations can...): Here Authors stated that the PY14⁺ cations can stabilize the polybromide and interhalogen anion species, but what is exactly the 'stabilization' mechanism? Recombination reaction? Solvation sheath formation? Coordination-bond formation? Is there any evidence of this? What do the UV spectra after 10 cycles have in relation to this and why the weakening of Br₀ peak is proof of this? If what Authors claimed is true (the two-step reaction of Br⁻/Br₀/Br⁺ is the main mechanism that helped to improve the Al reaction kinetics), isn't Br₀ the entity that should be abundant as it is the only entity that participates in both of the two-step reactions?

18. Also, any data of the Br-Cl compound that was stated in Introduction (page 4, line 71)?

19. Page 10, line 237-242: Why is it only by increasing the charge capacity then the reversible two-step Br⁻/Br₀/Br⁺ conversion was observed?

20. For Figure 4e, why do Authors decide to put the result of their main object, 3.25 m AlCl₃ + 1 m PY14Br in the Supporting Information (Supplementary Figure 20) and put the secondary one (3.25 m AlCl₃ + 1 m Et4NBr) in the main manuscript instead? Also, how can Authors say that both of these results are 'similar'? They are practically very different. Please elaborate on this.

21. For Figure 4f, it seems that the rate capability of the cell is not good because after 0.5 mA cm⁻², the initial voltage profile was not recovered well. How do Authors comment on this?

22. Please recheck all of the Figure captions, both in main manuscript and supporting information, and fix the captions accordingly. For example, please fix the caption of Supplementary Figure 22.

23. Page 12, line 287: Quantifying the Zn/Al ratio of the ZnAl alloy using EDS is not proper, while Authors also provided that the Al thickness was up to 12 micrometer (Figure 4b). Using other more proper quantification methods such as ICP-MS or ICP-OES is necessary (if Zn to Al ratio quantification is deemed important).

24. Page 12, line 307-309: What could be the reason for the similarity between + 1 m PY14Br and + 1 m Et4NBr that made both electrode performs 'similarly'?

25. How do Authors ensure that Zn dissolution did not occur and did not participate in the charge storage mechanism in the

full cell? If this was not verified beforehand, there is a high possibility that the superior energy and power metrics are not exclusive to the contribution of the two-step Br conversion reaction in the presence of Cl and make the whole story different.

26. Page 13, line 331: What do Authors mean by 'Even at 5 mA cm⁻² and 5 mAh cm⁻²,...'?

27. In general, this section is pretty ambiguous because it does not reveal the greatness of the designed electrolyte. Instead, it looks more like revealing the greatness of Zn-Al alloy in these systems. Also, why the + 1 m Et₄NBr electrolyte was not tested?

28. Page 13, line 340: The so-called 'conversion cathode' is just AC (activated carbon). Please be clearer in your statements.

29. Why do Authors make the anode 13x thicker than the cathode (278 μm anode vs. 20 μm cathode, Supplementary Figure 27)?

In summary, the authors should first build a systematic investigation and should only provide data-based arguments. Only after establishing strong and well-supported arguments based on robust data collection and proper analyses, this work can be considered meaningful.

Reviewer #2

(Remarks to the Author)

In this manuscript, a hydrate-melt electrolyte design utilizing cost-effective AlCl₃ and organic halide salts realizes the aqueous Al-Br battery devices with superior energy-power merit. The hydrate-melt feature significantly suppresses water activity and loosely bound halogen anions, thus expands the electrochemical stability. The electrolyte allows reversible Br⁻/Br⁰/Br⁺ conversion in cathodes with an ultra-high areal capacity, minimizes the polarization, and smooths the surface of Zn-Al anode. This work taps the potential in advanced aqueous aluminum-based batteries technology. I recommend it for publication to further upgrade the work after making the following major revisions:

1. The different bromine salts affect the GCD curve, especially the Coulomb efficiency. Thus, the selection principles of bromine salts should be summarized in the manuscript.
2. By simulating the solvation structure of Al³⁺ in AlCl₃-PY14Br electrolyte, the authors demonstrate that Al³⁺ has a strong ability to inhibit water activity. However, the effect of PY14Br cations couldn't be ignored and excluded on the inhibition of water activity and the solvation structure of Al³⁺ in the manuscript. The authors should supplement the relevant contents.
3. The corrosion of water on the anode is analyzed particularly in the manuscript. However, the shuttle effect of bromine is neglected. Whether are polybromides produced during electrochemical process and do they have a more serious corrosive effect on the alloy anode? The inhibition mechanism of hydrate-melt electrolyte toward the polybromide corrosion on alloy anode should be discussed.
4. In Figure 3, the polarization increases obviously with current density and cycle numbers. The influence of this phenomenon should be analyzed from the electrolyte effect on bromine species transformation in Br-Al full battery.
5. For Al-Br batteries, the relevant data (utilization ratio of active bromine in the electrolyte and so on) should be studied in detail to enrich the relevant contents.
6. In Figure 4f, the deposition/stripping behavior has not realized stable state in the first 20 h. The authors should extend the test at 5 mA cm⁻² before conducting other current density tests. Likewise, there is a significant hysteresis voltage in Figure 4g during the first 100 h. Specifically, additional characterization is needed to analyze whether those phenomena exist soft short circuit.

Reviewer #3

(Remarks to the Author)

The regulation of Al stripping/plating in aqueous environments, such as the inhibition of H₂ evolution and the anti-corrosion of Al anode surface, is determinative for the operation of Aluminium (Al)-based aqueous batteries (AABs). The authors employed cost-effective AlCl₃ and organic halide salts (PY14Br), to synthesize hydrate molten electrolytes. This approach effectively reduces the activity of H₂O within the electrolyte by facilitating strong ion associations between water molecules and loosely bound halogen anions, ultimately enhancing the electrochemical stability of the battery. This work is in general of novelty for the AAB field. However, some important issues have not been clearly clarified, which require further investigation for understanding the effect of the additive. This manuscript can be further considered after addressing those issues.

1. The authors have demonstrated through ab initio molecular dynamics (AIMD) simulations that over 98% of water molecules in the electrolyte contribute to the formation of the Al solvation structure. I would like to inquire whether the authors have validated the accuracy of the AIMD simulations through additional phase characterizations, such as spectral tests.
2. What is the molecular structure of the Al solvation complex that fully incorporates water molecules in the mixed electrolyte? Additionally, could you provide the corresponding experimental characterization?
3. To emphasize the structural advantages of PY14Br, it is recommended to include the results of the ²⁷Al nuclear magnetic resonance test for the AlCl₃ + Et₄NBr system.
4. Does the incorporation of organic PY14Br into the mixed electrolyte lead to a reduction in conductivity? Furthermore, given the importance of electrolyte additive stability for battery cycle longevity, it is recommended that the authors also assess the electrochemical stability of PY14Br throughout cycling.
5. The energy storage device assembled by the authors employs an Al/Zn alloy as the negative electrode. Have the authors investigated whether Zn participates in the electrochemical reaction and, if so, its contribution to capacity?
6. On page 4, line 82, what does 3.25 m mean? Is it mol L⁻¹?

Version 1:

Reviewer comments:

Reviewer #2

(Remarks to the Author)

The reply content meets most of the research problems, but the discussion on the selection principles of bromine salts is insufficient, it is necessary to provide a definitive data-oriented summary to explain the essential issue on the bromine salt in the term of different organic cations. And I would recommend the publication after further revision of the review.

Reviewer #3

(Remarks to the Author)

The authors have revised this paper carefully based on the reviewer's comments. I'd like to recommend its publication without alternation.

Reviewer #4

(Remarks to the Author)

The work titled "Hydrate-Melt Electrolyte Design for Aqueous Aluminium-1 Bromine Batteries with Superior Energy-Power Merits" presents the fabrication of AlCl_3 -based aqueous hydrate-melt electrolyte, which demonstrates good compatibility with both Zn/Al anodes and the $\text{Br}^-/\text{Br}_0/\text{Br}^+$ conversion. The unique double-sheath solvation structure of Al^{3+} revealed in the electrolyte enables both low water reactivity and a wide electrochemical stability window. The proposed electrolyte system is well characterized through a combination of coherent experimental and computational investigations. When using this electrolyte in Al-Br batteries, the designed electrolytes are endowed with high power density and high areal capacity up to 5 mAh cm^{-2} . In general, this manuscript is well organized, the high areal capacity for the cathode has demonstrated potential application interest. I have shown that in previous round of peer-review process, the three reviewers have raised insightful comments to improve the scientificity of manuscript, which have been mostly addressed by the authors. Therefore, in my opinion, the manuscript justifies publication in Nature Communications. I only have a few technical points needing further clarification by the authors.

1. The authors have attributed the great rate performance of the Al-Br battery to fast redox kinetics. To strengthen this claim, it would be helpful to include electrochemical impedance spectroscopy measurements to directly evaluate charge transfer characteristics. In addition, reporting the energy efficiency of the full cell at various current densities would provide valuable insight into the practical performance of the system under different operating conditions.
2. In practical applications, the operating temperature is not constant. Since the proposed hydrate-melt electrolyte relies on a highly structured solvation environment, it would be valuable to comment on whether this structure remains stable at different temperatures. If possible, the authors are encouraged to explore this aspect using AIMD or complementary experimental data.
3. The manuscript reports impressive capacity values enabled by the $\text{Br}^-/\text{Br}_0/\text{Br}^+$ conversion chemistry within an activated carbon host. To fully understand the contribution of the redox species, it would be helpful to clarify the intrinsic capacity of the activated carbon itself. Providing a control experiment to estimate baseline capacity from the carbon host alone would help quantify the actual contribution from the Br-based redox process.
4. The electrolyte exhibits excellent anodic stability, thereby enabling the reversible $\text{Br}^-/\text{Br}_0/\text{Br}^+$ conversion chemistry. In the manuscript, the anodic stability of the baseline electrolyte (3.25 m AlCl_3) is confirmed to extend beyond 1.5 V vs. Ag/AgCl. The authors then introduce the Br source in the form of organic bromide salts. It would be important to clarify whether the introduced organic cation affects the anodic stability. The authors are encouraged to perform LSV or potential floating measurements using 3.25 m AlCl_3 + 1 m PY14Cl to evaluate the influence of the organic cation.
5. The authors describe the proposed electrolyte as "cost-effective", which is a critical consideration for practical applications, particularly in large-scale energy storage. This claim would be more convincing if it is supported by comparative data. Therefore, it is recommended that the authors compare the cost of this electrolyte with other representative aqueous Al battery electrolytes, such as $\text{Al}(\text{OTf})_3$, of course taking consideration of salt concentration.

Version 2:

Reviewer comments:

Reviewer #2

(Remarks to the Author)

The authors have revised this paper carefully based on the reviewer's comments. I'd like to recommend its publication without alternation.

Reviewer #4

(Remarks to the Author)

The authors have addressed most of the concerns and improved the quality of the paper. I agree its publication on Nat.Commun.

To Reviewer 1:

The idea is somewhat interesting but not so much because there are many papers with similar approaches out there.

Response: We sincerely appreciate the reviewer's constructive comments. We are sorry for the lack of clarity in the original manuscript in convincing the novelty of this study. Actually, this work **introduces a conceptually new electrolyte design for Al-based aqueous batteries (AABs)**, specifically hydrate-melt electrolytes utilizing cost-effective AlCl_3 and organic halide salts. The optimal electrolyte features significantly suppressed water reactivity and loosely bound halogen anions, stemming from its unique electrolyte features: the majority of water molecules are tightly engaged in robust ion solvation, while halogen anions occupy the outer solvation sheath of cations. As an additional major contribution, **this study marks the first demonstration of $\text{Br}^-/\text{Br}^0/\text{Br}^+$ conversion for AABs, achieving an exceptional areal capacity of 5 mAh cm^{-2} , representing the state-of-the-art among all reported AAB cathodes (typically below 1 mAh cm^{-2}).** This outstanding performance is made possible by the distinctive features of our hydrate-melt electrolyte, ensuring excellent compatibility with the reversible $\text{Br}^-/\text{Br}^0/\text{Br}^+$ conversion. Furthermore, combining the $\text{Br}^-/\text{Br}^0/\text{Br}^+$ conversion cathode with the Zn-Al alloy anode yields aqueous Al-Br full cells with an average discharge voltage of 1.7 V. Thanks to the large areal capacity and rapid electrochemical kinetics of both electrodes, the full cells deliver **a volumetric energy density (267 Wh L^{-1}), comparable to commercial Li-ion batteries (e.g., 263 Wh L^{-1} , Panasonic CG420A) and a large volumetric power density (1069 W L^{-1}) at the level of electrochemical capacitors.** Aligned with the positive views expressed by other reviewers, we believe that the novelty, significance, and impact of this work justify its suitability for publication in *Nature Communications*.

The manuscript is also not well prepared for scientific publication due to some of the following reasons: 1) the scientific elaboration and arguments are lacking. 2) many of the claims in this work are overstated. 3) many of the claims are jumping into conclusions rather than stating data-based observation. These reasons reduce its proper readiness for scientific readership.

Response: We appreciate the reviewer for pointing out concerns regarding the manuscript clarity. According to your specific comments below, we have carried out additional experiments and expanded our discussions to ensure that our statements are well-supported and the manuscript is more reader-friendly with clear and detailed explanations. The specific actions taken to address these points are outlined in our point-by-point responses below and reflected in the revised manuscript.

Moreover, the use of 'melt electrolyte' term in the title is also misleading and looks more like a clickbait. Just because the organic halide salts being used in this work (KBr, PY14Br, Et_4NBr) were heated at

80°C to facilitate dissolution, which is also a relatively low temperature (< 100 °C), this ‘part’ of the electrolyte is not proper to be called as ‘melt’. They are just normal Br-based ionic liquids.

Response: We are sorry for any confusion caused regarding the term ‘hydrate-melt electrolyte’. In fact, this term has been widely used in the field of aqueous electrolytes to describe systems where the majority of water molecules are strongly associated with ions, leaving very few free water molecules.^{1,2} Therefore, we did not invent this term. It is important to note that the term is not defined by the electrolyte preparation procedure. In our study, we revealed that strongly associated water molecules account for > 98% of the total water content in our optimal 3.25 m AlCl₃ + 1 m PY14Br electrolyte. Thereby, our developed electrolyte aligns well with the concept of ‘hydrate-melt electrolyte’. Additionally, we would like to clarify that describing our system as ‘Br-based ionic liquids’ would be inaccurate due to the presence of water as a solvent. Ionic liquids are strictly defined as solvent-free liquids composed entirely of ions.

Action: We have incorporated this clarification into the revised manuscript. (Page 5, Paragraph 1)

This work attempted to offer a solution to increase the energy and power performance of Al-ion batteries in general, which is by introducing another ion into the hydrate AlCl₃ electrolyte system, by adding organic bromide salts. Authors stated that by adding Br into the system, the two-step Br conversion reaction (Br⁻/Br⁰/Br⁺) may result in “superb reaction kinetics that could potentially allow for a practical areal capacity, akin to metal stripping/plating reactions”. Authors stated that the first Br⁻/Br⁰ reaction step may “overcomes the hindrance associated with solid-state Al³⁺ diffusion during the charge storage process”. First of all, these two statements are not supported with the data presented by Authors.

Response: We appreciate the valuable comment of the reviewer. Indeed, the referred statements in the ‘Introduction’ section were not explicitly elaborated in the ‘Results’ section. However, these statements are clearly supported by our experimental results.

Specifically, we identified the interhalogen species involved in the conversion reaction through *in-situ* Raman spectroscopy (Fig. R1a). At the fully discharged state, the electrode shows almost no peaks, indicating that Br⁻ exists as a monatomic ion dissolved in the electrolyte. As the electrode is charged to 1.25 mAh cm⁻², characteristic peaks corresponding to Br₃⁻ (at 160 cm⁻¹) and Br_{2n+1}⁻ (at 205 cm⁻¹ and 252 cm⁻¹) emerge.³ Further charging of the electrode to 5 mAh cm⁻² reveals a new peak at 273 cm⁻¹, which corresponds to the symmetric stretching vibration of BrCl₂⁻.⁴ These results verify the conversion process from Br⁻ (a liquid species in the electrolyte) to Br_{2n+1}⁻/BrCl₂⁻ (confined within the activated

carbon host as solid), without involving solid-state Al^{3+} diffusion. This liquid-solid $\text{Br}^-/\text{Br}^0/\text{Br}^+$ conversion is analogous to the liquid-solid metal stripping/plating reactions.

Fig. R1 a *In situ* Raman spectra of the AC electrode during a charge process. **b** GCD profile of the AC electrode at 5 mA cm^{-2} in $3.25 \text{ m AlCl}_3 + 1 \text{ m PY14Br}$.

In addition, the rapid kinetics of the $\text{Br}^-/\text{Br}^0/\text{Br}^+$ conversion is evidenced by the small charge-discharge polarization observed in our electrochemical analysis (**Fig. R1b**). Even at an ultrahigh current density of 5 mA cm^{-2} , the charge/discharge profile of the electrode exhibits small potential polarizations of less than 100 mV. Moreover, our electrode achieves a practical areal capacity of 5 mAh cm^{-2} , which represents the highest value ever reported for AABs.

Action: To address the concern of the reviewer, we have included the relevant discussions into the revised manuscript. (**Page 9, Paragraph 3** and **Page 11, Paragraph 2**)

Furthermore, Authors also stated that “the second Br^0/Br^+ reaction tends to be initiated with the participation of Cl^- through interhalogen reactions, forming Br-Cl compounds” – which is also a statement not investigated nor supported with Authors’ data. There is no analytical data that presents the existence of any Br-Cl compound.

Response: We appreciate the valuable comment of the reviewer. This statement is well supported by the updated *in-situ* Raman spectra obtained for the AC electrode at various charge states, as shown in **Fig. R1a** (**Fig. 3c** in the manuscript). After charging the AC electrode to 5 mAh cm^{-2} , a Raman peak emerges at 273 cm^{-1} , which is characteristic of the symmetric stretching vibration of BrCl_2^- .⁴

Action: We have clarified this issue in the revised manuscript. (**Page 9, Paragraph 3**).

To construct the full cell, Authors also introduced Zn-Al alloy as the anode, which practically means that there is a possibility that Zn may also participate in the reaction – also something that Authors exclude in their investigation.

Response: To address this concern, we conducted additional experiments to confirm the negligible involvement of Zn. A detailed explanation can be found in our response to Q25 below.

In its current form, I think the investigation presented in this work lacks the necessary comprehensiveness and systematic approach. As such, I cannot recommend it for publication, particularly for the audience of Nature Communications. To address these issues, the authors should conduct a more thorough and in-depth study, providing clear explanations, detailed analyses, and further data-based scientific elaboration. Additionally, revising the writing to ensure it is more objective and data-driven is essential to meet the standards of scientific publication.

Response: We appreciate the constructive comments from the reviewer, which are highly valuable for further improving the quality of our manuscript. Accordingly, we have conducted additional experiments and discussions in the revised manuscript. Detailed point-by-point responses are provided below, and we hope that these revisions thoroughly address all the concerns raised.

Specific comments:

1. Page 6, line 126: Authors stated that “(Fig. 2b), ... indicates the compatibility of 3.25 m AlCl₃ with Br⁰/Br⁺ conversion chemistry. This is one example on how Authors are overstating their results and making claims not related to their data. The measurement setup in this case are very simple and standard (WE: Ti rod, RE: Ag/AgCl, CE: AC) with 3.25 m AlCl₃ as the electrolyte – no Br in the system. How can this prove this system’s compatibility with Br while it is not there?

Response: We are sorry for the confusion caused by our statement. In **Fig.2b**, we conducted a potential floating test to evaluate the anodic stability of 3.25 m AlCl₃ as the base electrolyte for further electrolyte design. The intention of the mentioned statement was to highlight that 3.25 m AlCl₃ enables a high electrochemical stability (up to *ca.* 1.7 V vs. SHE), sufficient to accommodate the Br⁰/Br⁺ conversion (1.3 V vs. SHE).

Action: To clarify this point, we have revised the statement, which can be found in the revised manuscript (**Page 7, Paragraph 1**), or as below.

‘It is notable that 3.25 m AlCl₃ shows a negligible leakage current density of 0.6 μA cm⁻² at a high potential of 1.5 V vs. Ag/AgCl (ca. 1.7 V vs. SHE) (Fig. 2b). This high anodic stability established 3.25 m AlCl₃ as a promising base electrolyte for further design to accommodate the Br⁰/Br⁺ conversion (1.3 V vs. SHE).’

2. Page 7, line 172: Authors stated that the addition of 1 m PY14Br breaks the hydrogen bonding network among water, contributing to suppressing water activity. Then how can double sheath of solvation layer majorly filled with water molecules (as Authors also presented in Figure 1) can be formed?

Response: We appreciate the valuable question of the reviewer. In the referred statement, our intention was to highlight the less bulky water molecule aggregates in 3.25 m AlCl₃ + 1 m PY14Br compared with 3.25 m AlCl₃ and 3.25 m AlCl₃ + 0.5 m KBr. This observation suggests a relatively more disrupted hydrogen-bonding network in 3.25 m AlCl₃ + 1 m PY14Br, which has been recognized as a key factor in suppressing water activity in aqueous electrolytes.⁵ Our statement does not conflict with the presence of a double water sheath around Al³⁺, as we did not imply that the hydrogen-bonding network in 3.25 m AlCl₃ + 1 m PY14Br is completely broken. The existence of double water sheath is supported by the *ab initio* molecular dynamics (AIMD) simulation analysis in 3.25 m AlCl₃, 3.25 m AlCl₃ + 0.5 m KBr, and 3.25 m AlCl₃ + 1 m PY14Br (Fig. 2e-g and Supplementary Fig. 13-15), and it can be attributed to the high charge density of Al³⁺, which enables Al³⁺ to strongly associate with two layers of polar water molecules.

Action: To avoid potential confusion, we have revised the corresponding discussion in the updated manuscript (Page 8, Paragraph 1), as shown below.

‘This observation indicates that the presence of less bulky water molecule aggregates in 3.25 m AlCl₃ + 1 m PY14Br compared with 3.25 m AlCl₃ and 3.25 m AlCl₃ + 0.5 m KBr. The relatively more disrupted hydrogen-bonding network in 3.25 m AlCl₃ + 1 m PY14Br contributes to suppressing water activity.’

3. Page 8, line 173: Authors stated ‘A similar conclusion can also be drawn (Supplementary Fig. 8)’, but did not elaborate further. Please at least make a summarizing sentence in the main manuscript or elaborate clearly in supporting information to make the data presentation meaningful.

Response: Thank you for the thoughtful suggestion.

Action: Accordingly, we have revised the corresponding discussion in the updated manuscript (Page 8, Paragraph 1), as shown below.

“A similar conclusion can be drawn by the comparison of the dielectric loss spectra for the three electrolytes (**Supplementary Fig. 10**). The spectrum of 3.25 m AlCl_3 + 1 m PY14Br exhibits a notably suppressed peak associated with bulk water aggregates at around 20 GHz, in contrast to those of 3.25 m AlCl_3 and 3.25 m AlCl_3 + 0.5 m KBr.”

4. Similarly, please elaborate further on Supplementary Figure 9 as well, which contains three sub-figures (9a-9c). What do Authors want to say or present by providing these snapshots? At least make some annotations or marks on what are the important parts of those images.

Response: The referred supplementary figure illustrates snapshots of the electrolyte structures after 5 ps in the AIMD simulations. While we agree that limited information can be directly derived from these snapshots due to the inclusion of hundreds of atoms, we believe their inclusion in the supplementary materials is essential and aligns with standard practices for AIMD analyses of electrolytes. These snapshots serve as the foundation for extracting further information, such as simulated wide-angle X-ray scattering (WAXS) spectra (**Supplementary Fig. 12**), radial distribution function (RDF) data (**Fig. 2e-g** and **Supplementary Fig. 13-15**) and typical ion solvation configurations (**Supplementary Fig. 17**).

Action: We have added the corresponding discussion in the updated Supplementary Materials (**Supplementary Figure 11**).

5. Similarly, please elaborate on Supplementary Figure 10, which contains six sub-figures (10a-10f). There must be some important points of these images so that the experiment was carried out that it has to be presented. Please use the ‘Supplementary Information’ field wisely and responsibly.

Response: We appreciate the thoughtful suggestion of the reviewer. The original **Supplementary Fig. 10** displayed radial distribution functions (RDFs) and corresponding integral curves extracted from AIMD using Visual Molecular Dynamics 1.9.3 (VMD 1.9.3), illustrating the atomic distance and distributions.

Action: To enhance clarity and support specific statements, we have included RDFs for K, N (from PY14⁺), Cl, and Br. Furthermore, the original **Supplementary Figure 10** has been divided into three separate figures (updated **Supplementary Fig. 13-15**) based on different electrolytes. Additional discussion related to these updated figures has been added to the revised Supplementary Materials (**Supplementary Figure 13-15**), as shown below.

Fig. R2 displays RDFs for 3.25 m AlCl_3 . Two prominent Al- O_{water} peaks are identified at 1.88 Å and 3.93 Å, indicative of the two solvation sheaths of Al^{3+} with the coordination number (CN) of 4.84 and 9.55 (**Fig. R2a**). The majority of Cl (64.1%) are found to occupy the outer solvation shell of Al^{3+} with an Al-Cl distance of 4.33 Å (**Fig. R2b**). A small fraction of Cl (17.9% estimated from the CN of 0.57) is located in the inner solvation shell of Al^{3+} with an Al-Cl distance of 2.23 Å. Additionally, Cl exhibits a single distinct solvation shell comprising O atoms from water at a distance of 3.08 Å with the CN of 4.97 (**Fig. R2c**). An Al-Al peak is observed at 8.98 Å with the CN of 6.17, indicating the presence of a long-range ordered structure within this electrolyte (**Fig. R2d**).

Fig. R2 RDFs (solid lines) and integral curves (dashed lines) extracted from AIMD simulations of 3.25 m AlCl_3 , including **a** Al- O_{water} , **b** Al-Cl, **c** Cl- O_{water} , and **d** Al-Al.

Fig. R3 displays RDFs for 3.25 m $\text{AlCl}_3 + 0.5$ m KBr. Two prominent Al- O_{water} peaks are identified at 1.88 Å and 3.98 Å, indicative of the two solvation sheaths of Al^{3+} with the coordination number (CN) of 4.69 and 8.52 (**Fig. R3a**). A significant fraction of Cl (61.5%) resides in the outer solvation sheath of Al^{3+} with an Al-Cl distance of 4.48 Å (**Fig. R3b**). A small fraction of Cl (15.4% estimated from the CN of 0.69) is located in the inner solvation sheath of Al^{3+} with an Al-Cl distance of 2.23 Å. Br only resides in the outer solvation of Al^{3+} with an Al-Br distance of 4.48 Å (**Fig. R3c**). Cl, Br, and K each exhibit

distinct solvation shells composed of O atoms from water: Cl-O_{water} at 3.08 Å with the CN of 5.05 (**Fig. R3d**), Br-O_{water} at 3.43 Å with the CN of 5.61 (**Fig. R3e**), and K-O_{water} at 2.88 Å with the CN of 2.25 (**Fig. R3f**). These solvation interactions contribute to further suppressing water activity. Finally, an Al-Al peak is observed at 7.98 Å with the CN of 5.97, indicating the existence of a long-range ordered structure within this electrolyte (**Fig. R3g**).

Fig. R3 RDFs (solid lines) and integral curves (dashed lines) extracted from AIMD simulations of 3.25 m AlCl₃ + 0.5 m KBr including **a** Al-O_{water}, **b** Al-Cl, **c** Al-Br, **d** Cl-O_{water}, **e** Br-O_{water}, **f** K-O_{water}, and **g** Al-Al.

Fig. R4 displays RDFs for 3.25 m AlCl₃ + 1 m PY14Br. As shown in **Fig. R4a-c**, Cl, Br, and N (from PY14⁺) each exhibit a distinct solvation shell composed of O atoms from water. These solvation shells are identified at 3.08 Å with the CN of 4.62 for Cl-O_{water}, at 3.23 Å with the CN of 4.98 for Br-O_{water}, and at 4.38 Å with the CN of 4.25 for N-O_{water}. These interactions effectively suppress water activity. Additionally, an Al-Al peak is observed at 8.70 Å with the CN of 6.19, indicating the presence of a long-range ordered structure in this electrolyte (**Fig. R4d**).

Fig. R4 RDFs (solid lines) and integral curves (dashed lines) extracted from AIMD simulations of 3.25 m AlCl₃ + 1 m PY14Br including **a** Cl-O_{water}, **b** Br-O_{water}, **c** N-O_{water}, and **d** Al-Al.

6. Similarly, please elaborate further on Supplementary Figure 11, which contains three sub-figures (11a-11c). What do Authors want to say or present by providing these snapshots? At least make some annotations or marks on what are the important parts of those images. One suggestion is by marking which area is the inner or outer solvation sheath.

Response: The referred supplementary figure illustrates the methodology used to calculate the water solvation energy of outer and inner solvation sheaths for Al³⁺ and the inner solvation sheath for Li⁺.

Action: Following your suggestion, we have expanded the discussion and revised the figure with additional annotations in the updated supplementary materials (**Supplementary Fig. 16**) to improve the clarity. Details can also be found below.

Fig. R5 DFT-calculated water solvation energy of **a** outer solvation sheath of Al³⁺, **b** inner solvation sheath of Al³⁺, and **c** inner solvation sheath of Li⁺.

The water solvation energies for the outer solvation sheath of Al³⁺ ($E_{Al-OuterWater}$), the inner solvation sheath of Al³⁺ ($E_{Al-InnerWater}$), and the inner solvation sheath for Li⁺ ($E_{Li-InnerWater}$) were calculated using the Vienna Ab initio Simulation Package (VASP) with the PBE exchange-correlation functional. For $E_{Al-OuterWater}$, we calculated the energy of Al³⁺ with six inner-sheath water molecules and eight outer-sheath water molecules ($E_{Al(InnerWater)_6(OuterWater)_8^{3+}}$), the energy of Al³⁺ with six inner-sheath water molecules and seven outer-sheath water molecules ($E_{Al(InnerWater)_6(OuterWater)_7^{3+}}$), and the energy of a free water molecule ($E_{FreeWater}$), as illustrated in **Fig. R5a**. Using **equation (R1)**, $E_{Al-OuterWater}$ was calculated to be -1.10 eV . For $E_{Al-InnerWater}$, we calculated the energy of Al³⁺ with six inner-sheath water

molecules ($E_{Al(InnerWater)_6^{3+}}$), the energy of Al^{3+} with five inner-sheath water molecules ($E_{Al(InnerWater)_5^{3+}}$), and the energy of a free water molecule ($E_{FreeWater}$), as illustrated in **Fig. R5b**. According to **equation (R2)**, $E_{Al-InnerWater}$ was calculated to be -3.22 eV. For $E_{Li-InnerWater}$, we calculated the energy of Li^+ with four inner-sheath water molecules ($E_{Li(InnerWater)_4^+}$), the energy of Li^+ with three inner-sheath water molecules ($E_{Li(InnerWater)_3^+}$), and the energy of a free water molecule ($E_{FreeWater}$), as illustrated in **Fig. R5c**. According to **equation (R3)**, $E_{Li-InnerWater}$ was calculated to be -1.02 eV.

$$E_{Al-OuterWater} = E_{Al(InnerWater)_6(OuterWater)_8^{3+}} - E_{Al(InnerWater)_6(OuterWater)_7^{3+}} - E_{FreeWater} \quad (R1)$$

$$E_{Al-InnerWater} = E_{Al(InnerWater)_6^{3+}} - E_{Al(InnerWater)_5^{3+}} - E_{FreeWater} \quad (R2)$$

$$E_{Li-InnerWater} = E_{Li(InnerWater)_4^+} - E_{Li(InnerWater)_3^+} - E_{FreeWater} \quad (R3)$$

7. Page 8 line 184: Authors stated that ‘... dominant Br^- (75%) and Cl^- (79.5%) anions are revealed to reside in the outer solvation sheath of Al^{3+} ’, but where is Br in Figure 2g? If adding Br was missed in the modeling, I think Authors need to repeat everything by including Br in all cases, not just by assuming Cl^- and Br^- as the same thing just because both are halide ions. Possible differences in terms of charge density, polarizability, ionic size, solubility, oxidizing power, etc, may affect the system. Even if Authors think that it will end up not having any difference, their similarity should be proven with data.

Fig. R6 RDFs (solid lines) and integral curves (dashed lines) of Al–Br extracted from AIMD simulations of 3.25 m $AlCl_3$ + 1 m PY14Br.

Response: Thank you for pointing out your concern regarding the original **Fig. 2g**. The figure indeed shows the structure derived from the simulation result of 3.25 m AlCl₃ + 1 m PY14Br. Due to the relative low concentration of Br⁻ (9.75 times less than Cl⁻), the Al-Br coordination number in the outer sheath of Al³⁺ was calculated to be only 0.27 (**Fig. R6**). This implies that Br⁻ anions are not necessarily present in the outer sheath of all Al³⁺ cations.

Action: To address your concern, we have replaced the original structure with another one derived from the simulation result of 3.25 m AlCl₃ + 1 m PY14Br (**Fig. R7**), which clearly shows the presence of both Cl⁻ and Br⁻. This updated structure is now included in the revised supplementary materials (**Supplementary Fig. 17**). Meanwhile, **Fig. 2g** has been replaced with **Fig. R6** to avoid any potential confusion.

Fig. R7 Representative Al³⁺ solvation structure extracted from the snapshot of an AIMD simulated cell for 3.25 m AlCl₃ + 1 m PY14Br.

8. Moreover, did Authors also include ‘temperature’ in the modeling process? Authors stated that the organic halide salts being used in this work (KBr, PY14Br, Et₄NBr) were heated at 80 °C to facilitate dissolution in AlCl₃. Was the elevated temperature just used for dissolution purposes? Are the electrolyte mixtures used at 80 °C during the whole electrochemical experiments or are they cooled down first before being used? If they are cooled down, how long after they are being cooled down do precipitates occur? Also, if they are considered at high concentrations, what about their viscosity?

Response: We appreciate the constructive questions of the reviewer. In the AIMD simulations, the temperature was set to room temperature (298.15 K), which is consistent with the conditions used for all our electrochemical measurements. The elevated temperature (80 °C) mentioned in the manuscript

was only applied during the electrolyte preparation to accelerate salt dissolution. After preparation, the electrolytes were stored and evaluated exclusively at room temperature. As shown in **Fig. R8**, all electrolytes remain transparent and free of precipitates after three months of storage at room temperature, demonstrating their long-term stability. Furthermore, we evaluated the viscosity of the electrolytes at 298.15 K through rotational viscometer. As shown in **Table R1**, the viscosity values are in the range of 38~58 mPa s, which is comparable with that of the classic highly concentrated electrolytes, such as 21 m LiTFSI (36 mPa s at 298.15 K) and 1 m LiTFSI + 7 m LiBETI (203 mPa s at 303.15 K).^{2,5}

Action: The relevant discussion has been included into the revised manuscript. (**Page 7, Paragraph 2**)

Fig. R8 Digital photos of our electrolytes after three months of storage at room temperature.

Table R1 Viscosities of our electrolytes at 298.15 K.

Electrolytes	Viscosity (mPa s)
3.25 m AlCl ₃	38
3.25 m AlCl ₃ + 0.5 m KBr	42
3.25 m AlCl ₃ + 1 m PY14Br	58
3.25 m AlCl ₃ + 1 m Et ₄ NBr	56

9. Page 8, line 190: Authors stated: ‘These loosely bound Br⁻ and Cl⁻ anions are expected to facilitate fast interfacial charge transfer during Br⁻/Br⁰/Br⁺ conversion by avoiding kinetics-limiting anion-cation dissociation.’ Is there any data that can support this claim?

Response: Thank you for the insightful comment. To support this statement, we conducted DFT calculations to evaluate the binding energies of Al³⁺-X⁻ and Al(H₂O)₆³⁺-X⁻ (X: Cl or Br). As shown in

Fig. R9, the binding energies of $\text{Al}^{3+}\text{-X}^-$ (-27.32 eV for $\text{Al}^{3+}\text{-Cl}^-$ and -28.20 eV for $\text{Al}^{3+}\text{-Br}^-$) are more than double those of $\text{Al}(\text{H}_2\text{O})_6^{3+}\text{-X}^-$ (-12.36 eV for $\text{Al}(\text{H}_2\text{O})_6^{3+}\text{-Cl}^-$ and -10.28 eV for $\text{Al}(\text{H}_2\text{O})_6^{3+}\text{-Br}^-$), indicating significantly faster Br^-/Cl^- dissociation kinetics in $\text{Al}(\text{H}_2\text{O})_6^{3+}\text{-X}^-$ than in $\text{Al}^{3+}\text{-X}^-$.

Action: The relevant discussion has been added into the revised manuscript. (**Page 9, Paragraph 1**)

Fig. R9 DFT-calculated binding energies of $\text{Al}^{3+}\text{-X}^-$ and $\text{Al}(\text{H}_2\text{O})_6^{3+}\text{-X}^-$ (X: Cl or Br).

10. Page 8, line 192: Authors stated: "...the statistical analysis of water molecules involved in strong solvation (including the two solvation sheaths of Al^{3+} and one solvation sheath of other ions) reveals minimal free water in 3.25 m AlCl_3 (6.1%), 3.25 m AlCl_3 + 0.5 m KBr (5.4%), and 3.25 m AlCl_3 + 1 m PY14Br (1.8%), underscoring the hydrate-melt nature of all three electrolytes.". How did Authors derive the statistical analysis and where is the data? A detailed description of how these are calculated and derived should be disclosed and discussed as well.

Table R2 Analysis of free water molecules in different electrolytes based on the AIMD result.

Electrolyte	Number of free water molecules	Total number of water molecules	Ratio of free water molecules
3.25 m AlCl_3	14	225	6.2%
3.25 m AlCl_3 + 0.5 m KBr	12	225	5.3%
3.25 m AlCl_3 + 1 m PY14Br	4	225	1.8%

Response: We conducted the statistical analysis using the Vienna Ab initio Simulation Package (VASP) based directly on the AIMD simulation results shown in **Supplementary Fig. 11**. Free water molecules were defined as those that are not present in either of the two solvation sheaths of Al^{3+} or in the single

solvation sheath of other ions (K^+ , $PY14^+$, Cl^- , and Br^-). In this sense, we set cutoff distances between water molecules and the various ions based on the RDF results (**Fig. 2e-g** and **Supplementary Fig. 13-15**, specifically, 5 Å for Al^{3+} , 5 Å for the N atom in $PY14^+$, 4 Å for K^+ , 4 Å for Cl^- , and 4 Å for Br^-). Using these criteria, we derived the number of free water molecules in different electrolytes (**Table R2**). The ratios of free water molecules in the electrolytes were determined by dividing the number of free water molecules by the total number of water molecules.

Action: The corresponding discussion has been provided in the revised manuscript. (**Page 9, Paragraph 1** and **Supplementary Table 2**)

11. Page 8, line 200: Authors stated that there are two pairs of redox peaks on the CV, but on the data, there are just three peaks (Figure 3a). One pair (two peaks) in the 0.6 – 0.7 region, which Authors assigned to Br^-/Br^0 conversion, while there is only one single peak between 0.8 – 1.0 V (vs. Ag/AgCl). Isn't this evidence of irreversibility of the Br^0/Br^+ conversion? Moreover, isn't the sudden steep current increment past 1.0 V indicative of OER?

Fig. R10 CV curve of the AC electrode in 3.25 m $AlCl_3$ + 1 m $PY14Br$ with a potential window of 0.20 ~ 1.16 V vs. Ag/AgCl at 0.1 mV s⁻¹.

Response: Thank you for your insightful questions. Regarding the second oxidation step at 0.20 ~ 1.10 V vs. Ag/AgCl in **Fig. 3a**, the increasing current is indicative of the occurrence of the Br^0/Br^+ conversion. We understand your concern regarding the absence of a distinct 'peak' shape with subsequent decreasing current. In fact, such a CV shape is quite common for electrochemical halogen conversion reactions.^{6,7} This behavior can be attributed to the cutoff potential set during the CV test, at which the Br^0/Br^+

conversion is still not limited by ion diffusion to the electrode surface. We further conducted an additional CV test with a slightly expanded potential window of 0.20 ~ 1.16 V vs. Ag/AgCl, as shown in **Fig. R10**. As expected, a current-decreasing step was observed for the Br⁰/Br⁺ oxidation. We would like to clarify that the CV shape in **Fig. 3a** does not indicate irreversibility of the Br⁰/Br⁺ conversion. Using **Fig. 3a**, we estimated the coulombic efficiency with the positive-value and negative-value areas, which reaches 98%. This coulombic efficiency demonstrates the high reversibility of the overall Br⁻/Br⁰/Br⁺ conversion. Additionally, the exclusion of OER is strongly supported by this close-to-unit coulombic efficiency and the high anodic stability of 3.25 m AlCl₃ (1.5 V vs. Ag/AgCl, **Fig. 2b**).

12. Page 8, line 202: Authors stated: “These two potential regions are consistent with the potentials of Br⁻/Br⁰ and Br⁰/Br⁺ conversion”. Please elaborate on this statement specifically and clearly, as to where each conversion reaction step occurs (similar to what Authors did on page 9 line 211-213). This is important now because the provided CV data does not support this claim well.

Response: As explained in our response to Q11, we believe that our CV data firmly supports our claim.

Action: According to your suggestion, we have modified the statement as below.

‘The redox peaks at the lower potential are attributed to the Br⁻/Br⁰ conversion based on the aligned potential, while the redox reaction at the higher potential corresponds to the Br⁰/Br⁺ conversion.’ (**Page 9, Paragraph 2**)

13. For Figure 3b, why are the peaks at 13470 eV not ‘smooth’ and looked more like a triangle (as if some data are missing, not high-resolution data), not like the peak beside it at around 13480 eV which is smooth?

Response: We appreciate the careful observation of the reviewer. The XANES spectrum data were collected from beamline P65 at Deutsches Elektronen-Synchrotron (DESY) with a standard data resolution set for the beamline (0.63 eV⁻¹). No data were manually removed from the raw data. The triangle-like shape observed for the sharp peak at 13470 eV may be attributed to resolution limit of the data. However, the intensities and positions of the peaks are sufficient to confidently confirm the generation of Br⁰ and Br⁺, as discussed in our manuscript.

14. For Figure 3c, I suggest Authors not only give dashed lines but also give annotations to provide better instructiveness of the data to the reader. What is the point of this image?

Response: Thank you for the valuable suggestion.

Action: Accordingly, **Fig. 3c** has been updated in the revised manuscript (as shown in **Fig. R11**) with more relevant discussion (**Page 9, Paragraph 3**). Details can also be found below.

‘Additionally, *in-situ* Raman spectra were collected in a home-made two-electrode electrochemical cell (**Supplementary Fig. 19**) to understand the interhalogen species during the conversion reaction (**Fig. R11**). At the fully discharged state, the electrode shows almost no peaks, as the Br^- is monatomic ion without Raman activity in the electrolyte. Along charging for 1.25 mAh cm^{-2} , the electrode starts to show characteristic peaks associated with Br_3^- (at 160 cm^{-1}) and Br_{2n+1}^- (at 205 cm^{-1} and 252 cm^{-1}).³ Moreover, charging the electrode for 5 mAh cm^{-2} induced the presence of a new peak at 273 cm^{-1} , a characteristic peak for the symmetric stretching vibration of BrCl_2^- .⁴

Fig. R11 *In situ* Raman spectra of the AC electrode during a charge process.

15. In Figure 3e, it seems like starting from 5 mA cm^{-2} to 20 mA cm^{-2} , there are significant iR drops during the charge-discharge test. How do Authors comment on this?

Response: It is natural that the IR drop increases with the applied current, since the IR drop is the product of the applied current and the ohmic resistance. We show a zoomed-in view of the IR drop region for the three discharge profiles in **Fig. R12**. The IR drop values were determined to be 13 mV, 34 mV, and 57 mV at 5 mA cm^{-2} , 10 mA cm^{-2} , and 20 mA cm^{-2} , respectively. In fact, considering the high current densities, these IR drop values are very small. For comparison, a typical NCM523 cathode for Li-ion batteries exhibited a IR drop of 100 mV even at a low current density of 0.5 mA cm^{-2} .⁸ The low IR drop in our system can be attributed to the efficient electron transport network formed by the AC, which ensures a low ohmic resistance.

Action: We have added the relevant discussion to the revised manuscript. (Page 11, Paragraph 2)

Fig. R12 Zoomed-in GCD profiles of the AC electrode at different current densities, highlighting the IR drop region.

16. At 1 C, the stability of the AC electrode reached 1500 cycles (Figure 3f), then why at 0.2 C the stability dropped to only 200 cycles (Supplementary Figure 15)?

Fig. R13 Cycling performance of the AC electrode in 3.25 m AlCl₃ + 1 m PY14Br at 1 mA cm⁻² and 5 mA cm⁻².

Response: We showed 200 cycles at 0.2 C due to the substantial time required for the measurement, not because of electrode degradation. At 0.2 C, each charge/discharge cycle takes 10 hours, and 200 cycles correspond to 2000 hours (over 83 days). We did not terminate the cycling test at 0.2 C. The updated

cycling performance (**Fig. R13**) reveals that the AC electrode remains stable even after 350 cycles at 0.2 C (approximately 146 days).

Action: This updated figure has been added to the revised manuscript (**Supplementary Fig. 23**).

17. Page 10, statement in Line 230-235 (In 3.25 m AlCl₃ + 1 m PY14Br, the PY14⁺ cations can...): Here Authors stated that the PY14⁺ cations can stabilize the polybromide and interhalogen anion species, but what is exactly the ‘stabilization’ mechanism? Recombination reaction? Solvation sheath formation? Coordination-bond formation? Is there any evidence of this? What do the UV spectra after 10 cycles have in relation to this and why the weakening of Br⁰ peak is proof of this? If what Authors claimed is true (the two-step reaction of Br⁻/Br⁰/Br⁺ is the main mechanism that helped to improve the Al reaction kinetics), isn’t Br⁰ the entity that should be abundant as it is the only entity that participates in both of the two-step reactions?

Fig. R14 Digital photos of **a** 3.25 m AlCl₃ + 1 m PY14Br and **b** 3.25 m AlCl₃ + 0.5 m KBr after the galvanostatic charge tests at 1 mA cm⁻² and 5 mAh cm⁻² using a three-electrode setup with a glass beaker.

Response: We notice that our original statement may have caused some misunderstanding. Our intention was to point out the capability of PY14⁺ cations to enhance the Br⁻/Br⁰/Br⁺ conversion reversibility by inhibiting the loss of active polyhalide species (e.g., Br_{2n+1}⁻ and BrCl₂⁻) from the AC electrode due to dissolution into the electrolyte. This stabilizing effect of bulky organic cations, such as quaternary ammonium and pyridinium cations, has been widely documented in previous reports.⁹⁻¹¹ These organic cations can electrostatically associate with anionic polyhalide species, forming water-insoluble phase that prevent their dissolution into the aqueous electrolyte. To validate this effect, we

performed galvanostatic charge test at 1 mA cm^{-2} and 5 mAh cm^{-2} using a three-electrode setup with a glass beaker. The electrolytes tested were $3.25 \text{ m AlCl}_3 + 1 \text{ m PY14Br}$ and $3.25 \text{ m AlCl}_3 + 0.5 \text{ m KBr}$. The setup comprised a Ti foil as the working electrode, an over-capacity AC electrode as the counter electrode, and an Ag/AgCl reference electrode. After charging, a water-insoluble phase formed in $3.25 \text{ m AlCl}_3 + 1 \text{ m PY14Br}$, confirming the effect of PY14^+ (**Fig. R14a**). In contrast, the generated anionic polyhalide species dissolved in $3.25 \text{ m AlCl}_3 + 0.5 \text{ m KBr}$, causing the electrolyte to turn yellow (**Fig. R14b**).

Action: To avoid confusion, we have revised the relevant description and included additional discussion in the updated manuscript. (**Page 10, Paragraph 1**)

Regarding the UV spectra mentioned by the reviewer (**Supplementary Fig. 20**), these measurements were conducted exclusively on the electrolyte extracted from the electrochemical cell after 10 galvanostatic charge/discharge cycles. The weakened Br^0 peak reflects the reduced dissolution of polybromide species into the electrolyte, further supporting the effect of both Et_4N^+ and PY14^+ in stabilizing polybromide species. During the $\text{Br}^-/\text{Br}^0/\text{Br}^+$ conversion, Br^0 species are dominantly confined within the electrode rather than being abundantly present in the electrolyte.

Action: The relevant explanation has been added to the revised manuscript. (**Supplementary Fig. 20**)

18. Also, any data of the Br-Cl compound that was stated in Introduction (page 4, line 71)?

Response: The presence of Br-Cl compound is well supported by the *in-situ* Raman spectra carried out for the AC electrode at different charge states, as shown in **Fig. R11 (Fig. 3c** in the manuscript). The detailed explanation can be found in our response to Q14.

19. Page 10, line 237-242: Why is it only by increasing the charge capacity then the reversible two-step $\text{Br}^-/\text{Br}^0/\text{Br}^+$ conversion was observed?

Response: We appreciate the insightful question of the reviewer. Ideally, in an electrochemical liquid-solid conversion process, the reaction potential remains consistent as long as the electrode/electrolyte interface remains relatively stable, until the full depletion of the reactant. However, in realistic cases, the electrochemical reaction alters the surface chemistry of the electrode, which in turn induces increasing electrochemical polarization. Specifically, during the electrode charging in our case (**Fig. 3d**), the electrode potential quickly reaches the threshold for initiating Br^-/Br^0 conversion. As the reaction progresses, the generated polybromide species continuously modify the electrode/electrolyte interface. This modification leads to a gradual increase in potential polarization, causing the electrode potential to

rise progressively throughout the Br^-/Br^0 conversion. By increasing the charge capacity, the electrode potential eventually surpasses the threshold required to initiate the Br^0/Br^+ conversion. At this point, the two-step $\text{Br}^-/\text{Br}^0/\text{Br}^+$ conversion is observed.

Action: We have included the relevant discussion into the revised manuscript. (Page 11, Paragraph 1)

20. For Figure 4e, why do Authors decide to put the result of their main object, 3.25 m AlCl_3 + 1 m PY14Br in the Supporting Information (Supplementary Figure 20) and put the secondary one (3.25 m AlCl_3 + 1 m Et_4NBr) in the main manuscript instead? Also, how can Authors say that both of these results are ‘similar’? They are practically very different. Please elaborate on this.

Response: Thank you for pointing out the concern regarding the figure arrangement. The reason we put the result of 3.25 m AlCl_3 + 1 m PY14Br in the Supplementary Materials (Supplementary Fig. 27) rather than the main manuscript is due to the relatively high level of noise observed in its signal. This noise likely stems from the instability of the PY14^+ cations under laser exposure during the sum frequency generation (SFG) measurement. In contrast, 3.25 m AlCl_3 + 1 m Et_4NBr has relatively high-quality SFG signals, as the Et_4N^+ cations have greater stability under laser exposure. The SFG result of 3.25 m AlCl_3 + 1 m Et_4NBr clearly demonstrates the suppression of hydrogen evolution as the potential decreases, making it more suitable for inclusion in the main manuscript (Fig. 4e). When we stated that ‘Similar suppression was observed for 3.25 m AlCl_3 + 1 m PY14Br’, we were referring to the consistent trend of hydrogen evolution suppression in both systems, despite the apparent noise in the SFG signal of 3.25 m AlCl_3 + 1 m PY14Br. The stable SFG signal along the potential decrease for 3.25 m AlCl_3 + 1 m PY14Br still supports this conclusion.

Action: To avoid any potential confusion, we have revised the relevant description in the updated manuscript. (Page 14, Paragraph 1)

21. For Figure 4f, it seems that the rate capability of the cell is not good because after 0.5 mA cm^{-2} , the initial voltage profile was not recovered well. How do Authors comment on this?

Response: We appreciate the valuable question of the reviewer. Fig. 4f shows the galvanostatic stripping/plating of the Zn-Al//Zn-Al symmetric cell at different current densities. In fact, the smaller voltage reflects the lower stripping/plating overpotential, which indicates better stripping/plating performance. In the initial 20 hours, the symmetric cell undergoes an interfacial activation step, a common phenomenon in such galvanostatic stripping/plating measurements. This activation step results in a gradual decrease in overpotential. Clearly, the interfacial activation was not fully completed during the initial 20 hours and continued during subsequent testing at smaller current densities. This explains

why the later profiles at 5 mAh cm^{-2} exhibit lower overpotential compared to the initial profiles at the same current density.

Fig. R15. Galvanostatic stripping/plating of the Zn-Al//Zn-Al symmetric cell in $3.25 \text{ m AlCl}_3 + 1 \text{ m PY14Br}$ **a** initially at 5 mA cm^{-2} for 100 h and **b** subsequently at different current densities.

To further address your concern, we conducted additional galvanostatic stripping/plating measurements of the Zn-Al//Zn-Al symmetric cell at 5 mAh cm^{-2} for an extended duration of 100 hours to ensure the completion of the interfacial activation step (**Fig. R15a**). Subsequently, we repeated the rate measurement as shown in **Fig. 4f**. As shown in **Fig. R15b**, the cell demonstrated nearly identical overpotential between the initial and later profiles at 5 mAh cm^{-2} .

Action: The relevant discussion has been added into the revised manuscript. (**Page 14, Paragraph 2**)

22. Please recheck all of the Figure captions, both in main manuscript and supporting information, and fix the captions accordingly. For example, please fix the caption of Supplementary Figure 22.

Response: Thank you for the kind reminder. We have thoroughly double-checked all the figure captions to ensure their accuracy. Upon review, we found no errors in the caption of the referred supplementary figure (**Supplementary Fig. 30**).

23. Page 12, line 287: Quantifying the Zn/Al ratio of the ZnAl alloy using EDS is not proper, while Authors also provided that the Al thickness was up to 12 micrometer (Figure 4b). Using other more proper quantification methods such as ICP-MS or ICP-OES is necessary (if Zn to Al ratio quantification is deemed important).

Response: According to your valuable suggestion, we have carried out the ICP-OES measurement of the Zn-Al alloy by dissolving it in 0.1 M HNO₃. The quantification result is shown in **Table R3**, and the Zn/Al atomic ratio was confirmed to be 0.17.

Action: The relevant discussion has been added to the revised manuscript. (**Page 13, Paragraph 1**)

Table R3 Elemental quantification of the Zn-Al alloy by ICP-OES.

Elements	Mass percentage (%)	Atomic percentage (%)
Zn	30.1	14.9
Al	69.9	85.1

24. Page 12, line 307-309: What could be the reason for the similarity between + 1 m PY14Br and + 1 m Et₄NBr that made both electrodes perform ‘similarly’?

Response: The similarity between + 1 m PY14Br and + 1 m Et₄NBr could be attributed to the bulky organic nature of PY14⁺ and Et₄N⁺ cations. At the anode side, these cations could preferentially adsorb onto the anode surface, thereby inhibiting hydrogen evolution. At the cathode side, as explained in our response to Q17, these cations are capable of associating with polyhalide species, effectively suppressing their dissolution into the aqueous electrolyte.

25. How do Authors ensure that Zn dissolution did not occur and did not participate in the charge storage mechanism in the full cell? If this was not verified beforehand, there is a high possibility that the superior energy and power metrics are not exclusive to the contribution of the two-step Br conversion reaction in the presence of Cl and make the whole story different.

Response: We appreciate the constructive question of the reviewer. To assess the dissolution of Zn, we first conducted a metal stripping experiment for the Zn-Al alloy electrode in 3.25 m AlCl₃ + 1 m PY14Br at 5 mA cm⁻² for 1 h, using a three-electrode setup comprising a Zn-Al alloy working electrode with an area of 1 cm⁻², an over-capacity AC electrode as the counter electrode, and an Ag/AgCl reference electrode. Afterward, the electrolyte was extracted from the cell and analyzed via ICP-OES to quantify the stripped Zn in the electrolyte. The stripped Zn was determined to be 3.6 × 10⁻⁶ mol, corresponding to a capacity of 0.195 mAh. Based on this analysis, we conclude that the participation of Zn in the process is minimal, accounting for only 3.9 % of the total Zn-Al alloy capacity. To assess the participation of Zn in the full cell, we employed ICP-OES to explore the electrolyte in Al-Br cell after 10 cycles at 5 mA cm⁻² and 5 mAh cm⁻². The calculated Zn concentration in the cycled electrolyte is negligible, only 7.5 × 10⁻³ m. Therefore, Zn shouldn't affect the electrochemical behavior of Br in the full battery.

Action: The corresponding discussion has been added to the revised manuscript. (Page 14, Paragraph 3 and Page 15, Paragraph 1)

26. Page 13, line 331: What do Authors mean by 'Even at 5 mA cm⁻² and 5 mAh cm⁻²,...'?

Response: Large current densities like 5 mA cm⁻², combined with large areal capacities like 5 mAh cm⁻², are considered harsh operating conditions for battery electrodes, particularly metal anodes.^{12,13} These conditions often result in significant electrode overpotential and accelerated degradation due to the amplified side reactions. However, achieving stability under such conditions is crucial for practical applications, as they enable fast-charging capabilities and high energy densities in devices. With the referred statement, we aim to emphasize the electrode's ability to maintain stable operation, characterized by low overpotential and long-term stability, even under these demanding conditions.

Action: The relevant explanation has been added to the revised manuscript. (Page 14, Paragraph 2)

27. In general, this section is pretty ambiguous because it does not reveal the greatness of the designed electrolyte. Instead, it looks more like revealing the greatness of Zn-Al alloy in these systems. Also, why the + 1 m Et₄NBr electrolyte was not tested?

Response: We are sorry for the confusion caused regarding the Zn-Al alloy anode evaluation section. As the development of battery electrolytes must consider compatibility with both the cathode and anode, we believe evaluating the specific anode performance in our developed electrolytes is of critical importance. In this section, we confirmed the good compatibility of our electrolyte with the Zn-Al alloy anode and elucidated the key roles of bulky organic cations in inhibiting hydrogen evolution at the anode

surface. These insights are crucial for guiding future electrolyte designs and lay the foundation for subsequent full-cell assembly. Since 3.25 m AlCl₃ + 1 m PY14Br was identified optimal for the AC cathode in facilitating the Br⁻/Br⁰/Br⁺ conversion (**Fig. 3**), we primarily focused on this electrolyte for the Zn-Al alloy anode evaluation. However, we should mention that we also provided key evaluations of 3.25 m AlCl₃ + 1 m Et₄NBr for the Zn-Al alloy anode, including Zn-Al stripping/plating measurements (**Supplementary Fig. 26**) and the SFG analysis (**Fig. 4e**), ensuring a comprehensive discussion of its performance. (**Page 13, Paragraph 3-4**)

28. Page 13, line 340: The so-called ‘conversion cathode’ is just AC (activated carbon). Please be clearer in your statements.

Response: We appreciate the reviewer for pointing out this potential confusion.

Action: Accordingly, we have revised the corresponding statement in the revised manuscript (**Page 15, Paragraph 1**), as shown below.

‘Given the desirable compatibility of the 3.25 m AlCl₃ + 1 m PY14Br electrolyte with both Zn-Al alloy anode chemistry and the Br⁻/Br⁰/Br⁺ conversion cathode chemistry, we further assembled aqueous Al-Br full cells using two-electrode Swagelok cells.’

29. Why do Authors make the anode 13x thicker than the cathode (278 μm anode vs. 20 μm cathode, Supplementary Figure 27)?

Response: Thank you for the insightful question. In fact, we employed AC cathodes with a thickness of 278 μm and Zn-Al alloy anodes with a thickness of 20 μm for full cell assembly. In practical batteries with metal anodes, it is typical for the cathode to be significantly thicker than the anode, primarily due to the high volumetric capacity of the metal anode. This ensures a balanced areal capacity between the anode and cathode. In our case, we confirmed the 278 μm cathode could accommodate reversible Br⁻/Br⁰/Br⁺ conversion with a high areal capacity of 5 mAh cm⁻² (**Fig. 3e**). For the anode, we used a 20 μm Zn-Al alloy anode, which was prepared by the thinnest Zn foil available in our lab, and confirmed its ability to achieve an areal capacity of 5 mAh cm⁻² (**Supplementary Fig. 30**), matching the areal capacity of the 278 μm cathode. Thereby, we assembled the full device with the 278 μm cathode and the 20 μm Zn-Al anode. To further improve the volumetric performance of the full cell, it would be beneficial to pursue a thicker cathode to further enhance areal capacity or explore thinner anodes to reduce the overall cell volume.

Action: The relevant discussion has been added to the revised manuscript. (**Page 15, Paragraph 1**)

In summary, the authors should first build a systematic investigation and should only provide data-based arguments. Only after establishing strong and well-supported arguments based on robust data collection and proper analyses, this work can be considered meaningful.

Response: We appreciate the valuable and insightful feedback from the reviewer. Accordingly, we have conducted additional experiments and expanded our discussions to thoroughly address all the concerns raised. We believe these revisions have helped clarify the novelty and significance of our study.

To Reviewer 2:

In this manuscript, a hydrate-melt electrolyte design utilizing cost-effective AlCl_3 and organic halide salts realizes the aqueous Al-Br battery devices with superior energy-power merit. The hydrate-melt feature significantly suppresses water activity and loosely bound halogen anions, thus expands the electrochemical stability. The electrolyte allows reversible $\text{Br}^-/\text{Br}^0/\text{Br}^+$ conversion in cathodes with an ultra-high areal capacity, minimizes the polarization, and smooths the surface of Zn-Al anode. This work taps the potential in advanced aqueous aluminum-based batteries technology. I recommend it for publication to further upgrade the work after making the following major revisions:

Response: We appreciate the positive comment of the reviewer. Additional experiments and discussions have been conducted to address the following concerns.

1. The different bromine salts affect the GCD curve, especially the Coulomb efficiency. Thus, the selection principles of bromine salts should be summarized in the manuscript.

Fig. R16 Digital photos of **a** $3.25 \text{ m AlCl}_3 + 1 \text{ m PY14Br}$ and **b** $3.25 \text{ m AlCl}_3 + 0.5 \text{ m KBr}$ after the galvanostatic charge tests at 1 mA cm^{-2} and 5 mAh cm^{-2} using a three-electrode setup with a glass beaker.

Response: Thank you for the valuable suggestion. Our study indicates that Br salts with bulky organic cations (like PY14^+ and Et_4N^+) are essential to enhance the $\text{Br}^-/\text{Br}^0/\text{Br}^+$ conversion reversibility. These organic cations can electrostatically associate with anionic polyhalide species (e.g., Br_{2n+1}^- and BrCl_2^-), forming water-insoluble phase that prevent their dissolution into the aqueous electrolyte. To validate this effect, we performed galvanostatic charge test at 1 mA cm^{-2} and 5 mAh cm^{-2} using a three-electrode setup with a glass beaker. The electrolytes tested were $3.25 \text{ m AlCl}_3 + 1 \text{ m PY14Br}$ and $3.25 \text{ m AlCl}_3 +$

0.5 m KBr. The setup comprised a Ti foil as the working electrode, an over-capacity AC electrode as the counter electrode, and an Ag/AgCl reference electrode. After charging, a water-insoluble phase formed in 3.25 m AlCl₃ + 1 m PY14Br, confirming the effect of PY14⁺ (**Fig. R16a**). In contrast, the generated anionic polyhalide species dissolved in 3.25 m AlCl₃ + 0.5 m KBr, causing the electrolyte to turn yellow (**Fig. R16b**).

Action: We have included the relevant discussion in the revised manuscript. (**Page 10, Paragraph 1**)

2. By simulating the solvation structure of Al³⁺ in AlCl₃-PY14Br electrolyte, the authors demonstrate that Al³⁺ has a strong ability to inhibit water activity. However, the effect of PY14Br cations couldn't be ignored and excluded on the inhibition of water activity and the solvation structure of Al³⁺ in the manuscript. The authors should supplement the relevant contents.

Fig. R17 RDFs (solid lines) and integral curves (dashed lines) extracted from AIMD simulations of 3.25 m AlCl₃ + 1 m PY14Br including **a** Br-O_{water} and **b** N-O_{water}.

Response: Indeed, the addition of PY14Br further inhibits water activity, as evidenced by the wide-angle X-ray scattering (WAXS, **Fig. 2d**) and dielectric relaxation spectra (**Supplementary Fig. 10**) of 3.25 m AlCl₃ and 3.25 m AlCl₃ + 1 m PY14Br. This reduction in water activity is attributed to the decrease in free water molecules in 3.25 m AlCl₃ + 1 m PY14Br (1.8%), in comparison with 3.25 m AlCl₃ (6.1%), as indicated by the *ab initio* molecular dynamics (AIMD) simulations. The simulated radial distribution functions (RDFs) of 3.25 m AlCl₃ + 1 m PY14Br provide crucial insights into the distribution of O (from H₂O), around Br (**Fig. R17a**) and N (from PY14⁺, **Fig. R17b**). Specifically, the Br-O_{water} peak at 3.23 Å with a coordination number (CN) of 4.98 and the N-O_{water} peak at 4.38 Å with a CN of 4.25 indicate that both Br⁻ and PY14⁺ contribute to the association with water molecules, thereby restricting water activity.

Fig. R18 RDFs (solid lines) and integral curves (dashed lines) of Al–O_{water} extracted from AIMD simulations of **a** 3.25 m AlCl₃ and **b** 3.25 m AlCl₃ + 1 m PY14Br.

To understand the impact of PY14Br on the solvation structure of Al³⁺, we compared the distribution of O (from H₂O) around Al in the simulated RDFs of 3.25 m AlCl₃ (**Fig. R18a**) and 3.25 m AlCl₃ + 1 m PY14Br (**Fig. R18b**). Both electrolytes exhibit similar characteristics in the inner solvation sheath (1.88 Å with a CN of 4.84 for 3.25 m AlCl₃ vs. 1.88 Å with a CN of 4.90 for 3.25 m AlCl₃ + 1 m PY14Br) and outer solvation sheath (3.93 Å with a CN of 9.55 vs. 4.03 Å with a CN of 9.21) of Al³⁺. Therefore, we can conclude that PY14Br does not apparently affect the solvation structure of Al³⁺.

Action: The corresponding discussion has been added to the revised manuscript. (**Page 7, Paragraph 3; Page 8, Paragraph 1-3 and Page 9, Paragraph 1**)

3. The corrosion of water on the anode is analyzed particularly in the manuscript. However, the shuttle effect of bromine is neglected. Whether are polybromides produced during electrochemical process and do they have a more serious corrosive effect on the alloy anode? The inhibition mechanism of hydrate-melt electrolyte toward the polybromide corrosion on alloy anode should be discussed.

Response: We appreciate the constructive suggestion from the reviewer. Indeed, oxidative polyhalide species (e.g., Br_{2n+1}⁻ and BrCl₂⁻) are known to be corrosive toward metals like Zn and Al.¹⁴ To specifically evaluate the issue of polyhalide corrosion while excluding other factors influencing the Zn-Al alloy morphology, we conducted galvanostatic charge-discharge tests at 5 mA cm⁻² and 5 mAh cm⁻² for 10 cycles. These tests were performed using a three-electrode setup comprising an AC electrode as the hosting electrode, an over-capacity AC electrode as the counter electrode, and an Ag/AgCl reference electrode. Three different electrolytes were employed, including 3.25 m AlCl₃ + 0.5 m KBr, 3.25 m AlCl₃ + 1 m PY14Br, and 3.25 m AlCl₃ + 1 m Et₄NBr. After the charge-discharge tests, the cycled electrolytes were extracted from the electrochemical cells, and a fresh Zn-Al alloy foil (**Fig. R19a**) was

immersed in each electrolyte for 1 hour. Severe corrosion, evidenced by dense surface holes, was observed on the foil immersed in 3.25 m AlCl₃ + 0.5 m KBr (**Fig. R19b**). In contrast, the corrosion issue was significantly mitigated for the foils immersed in 3.25 m AlCl₃ + 1 m PY14Br (**Fig. R19c**) and 3.25 m AlCl₃ + 1 m Et₄NBr (**Fig. R19d**).

Fig. R19 SEM images of **a** the fresh Zn-Al alloy foil, and the Zn-Al alloy foils after immersion in the cycled **b** 3.25 m AlCl₃ + 0.5 m KBr, **c** 3.25 m AlCl₃ + 1 m PY14Br, and **d** 3.25 m AlCl₃ + 1 m Et₄NBr electrolytes.

As explained in our response to Q1, bulky organic cations like PY14⁺ and Et₄N⁺ assist in stabilizing the generated polyhalide species by confining them predominantly within the AC cathode. This stabilization mechanism reduces the shuttling of polyhalide species to the anode, which can otherwise cause significant corrosion. The high coulombic efficiencies of the AC cathode in 3.25 m AlCl₃ + 1 m PY14Br (99.6%) and 3.25 m AlCl₃ + 1 m Et₄NBr (98.6%) further support this conclusion. As only a limited amount of polyhalide species shuttle to the anode in these electrolytes, the corrosion of the alloy anode is effectively inhibited.

Action: The relevant discussion has been added to the revised manuscript. (**Page 15, Paragraph 1**)

4. In Figure 3, the polarization increases obviously with current density and cycle numbers. The influence of this phenomenon should be analyzed from the electrolyte effect on bromine species transformation in Br-Al full battery.

Fig. R20 **a** GCD profiles of the AC electrode at different current densities with a capacity of 5 mAh cm⁻² as the charge cutoff in three-electrode cell. **b** GCD profiles of Al-Br full cell at different current densities with a capacity of 5 mAh cm⁻² as the charge cutoff. **c** GCD profiles of the AC electrode at different cycles during the cycling test at 5 mA cm⁻² in three-electrode cell. **d** GCD profiles of Al-Br full cell at different cycles during the cycling test at 5 mA cm⁻².

Response: Thank you for the valuable suggestion. To facilitate comparison between the three-electrode test and Al-Br full cell test, we have replotted the relevant GCD curves at different current densities and cycle numbers (**Fig. R20a-b**). We took the mid-capacity polarization as the standard for comparison. In the three-electrode test (**Fig. R20a**), the AC electrode shows a modestly increasing polarization along with the current density increase (90 mV at 5 mA cm⁻², 127 mV at 10 mA cm⁻², and 202 mV at 20 mA cm⁻²), reflecting the fast Br⁻/Br⁰/Br⁺ conversion kinetics. In comparison, the Al-Br full cell (**Fig. R20b**) shows slightly higher polarization than the three-electrode test at the same current densities (135 mV at

5 mA cm⁻², 176 mV at 10 mA cm⁻², and 300 mV at 20 mA cm⁻²). This polarization enlargement is attributed to the additional overpotential from the Zn-Al alloy anode.

In the three-electrode test at 5 mA cm⁻² (**Fig. R20c**), the AC electrode initially shows a slight decrease in polarization (from 115 mV at the 10th cycle to 90 mV at the 50th cycles) and subsequently maintains a constant polarization during cycling (90 mV at the 200th cycles), reflecting the superior stability of the Br⁻/Br⁰/Br⁺ conversion chemistry. The initial decrease in polarization can be attributed to the activation process of the electrode, which improves the electrolyte wettability of the electrode. For the Al-Br full cell at 5 mA cm⁻² (**Fig. R20d**), the polarization initially decreases and then increases with cycling (161 mV at the 10th cycle, 141 mV at the 50th cycle, 149 mV at the 100th cycle, 157 mV at the 150th cycle, and 170 mV at the 200th cycle). The increasing overpotential is likely due to the enlarged electrode resistance as cycling progresses.

Action: The corresponding discussion has been added to the revised manuscript. (**Page 11, Paragraph 2-3; Page 15, Paragraph 2 and Page 16, Paragraph 2**)

5. For Al-Br batteries, the relevant data (utilization ratio of active bromine in the electrolyte and so on) should be studied in detail to enrich the relevant contents.

Response: Following your suggestion, we evaluated the Br utilization efficiency (10~50%) for the Al-Br full cell by controlling the electrolyte amount (**Fig. R21a**). For Br utilization efficiencies ranging from 10% to 40%, the Al-Br cell exhibits nearly identical GCD curves with two distinct charge/discharge plateaus, indicating no significant side effects on the cell performance as the Br utilization efficiency increases within this range. When the Br utilization efficiency reaches 50%, the Al-Br cell shows an obvious performance deterioration. Moreover, we studied the temperature-dependent performance of the Al-Br cell (**Fig. R21b**). Along the temperature increases, the discharge capacity of the Al-Br cell gradually decreases (4.88 mAh cm⁻² at 298.15 K, 4.85 mAh cm⁻² at 303.15 K, 4.73 mAh cm⁻² at 313.15 K, 4.65 mAh cm⁻² at 323.15 K, and 4.42 mAh cm⁻² at 333.15 K). This capacity decay can be attributed to the accelerated dissolution of active Br species at elevated temperatures.

Action: We have included the relevant discussion in the revised manuscript. (**Page 15, Paragraph 1 and Page 17, Paragraph 1**)

Fig. R21 a GCD profiles of the Al-Br full cells at different Br utilization efficiencies at 5 mA cm^{-2} with a capacity of 5 mAh cm^{-2} as the charge cutoff. **b** GCD profiles of the Al-Br full cells at different temperature at 5 mA cm^{-2} with a capacity of 5 mAh cm^{-2} as the charge cutoff.

6. In Figure 4f, the deposition/stripping behavior has not realized stable state in the first 20 h. The authors should extend the test at 5 mA cm^{-2} before conducting other current density tests. Likewise, there is a significant hysteresis voltage in Figure 4g during the first 100 h. Specifically, additional characterization is needed to analyze whether those phenomena exist soft short circuit.

Response: We appreciate the insightful suggestions from the reviewer. Accordingly, we conducted additional galvanostatic stripping/plating measurements of the Zn-Al//Zn-Al symmetric cell at 5 mAh cm^{-2} for an extended duration of 100 hours to ensure the completion of the interfacial activation step (**Fig. R22a**). Subsequently, we repeated the rate measurement conducted for **Fig. 4f**. As shown in **Fig. R22b**, the cell demonstrated nearly identical overpotential between the initial and later profiles at 5 mAh cm^{-2} , confirming the stability after interfacial activation.

Fig. R22 Galvanostatic stripping/plating of the Zn-Al//Zn-Al symmetric cell in 3.25 m AlCl₃ + 1 m PY14Br at **a** 5 mA cm⁻² for 100 h and **b** different current densities subsequently.

To assess the potential issue of soft short circuiting, we assembled a Zn-Al//Zn-Al symmetric cell using 3.25 m AlCl₃ + 1 m PY14Br, and galvanostatic stripping/plating measurements were performed at 5 mA cm⁻² and 5 mAh cm⁻² for 100 hours at 298.15 K as the activation step. Afterward, temperature-dependent galvanostatic stripping/plating tests were conducted under the same current density and capacity conditions. The activation energy (E_a) for the cell reaction was determined by fitting the overpotential data in **Fig. R23** using **equation (R4)**, where T represents the temperature and η represents the overpotential. The calculated E_a value of 18.4 kJ mol⁻² is comparable to that of other metal stripping/plating reactions, such as ~20 kJ mol⁻² for Zn anode, and is notably greater than zero. Since metal stripping/plating reaction kinetics positively correlate with temperature ($E_a > 0$), whereas electrical conductivity exhibits the opposite trend ($E_a < 0$),¹⁵ these results collectively confirm the absence of soft short circuiting.

$$E_a = - \frac{T}{1000 \ln(\eta)} \quad (\mathbf{R4})$$

Fig. R23 **a** Temperature dependence of the overpotential for the Zn-Al//Zn-Al symmetric cell in 3.25 m AlCl₃ + 1 m PY14Br at 5 mA cm⁻² and **b** the corresponding Arrhenius plot, revealing an activation energy of 18.4 kJ mol⁻¹.

Action: The relevant discussion has been added to the revised manuscript. (**Page 14, Paragraph 2**)

To Reviewer 3:

The regulation of Al stripping/plating in aqueous environments, such as the inhibition of H₂ evolution and the anti-corrosion of Al anode surface, is determinative for the operation of Aluminium (Al)-based aqueous batteries (AABs). The authors employed cost-effective AlCl₃ and organic halide salts (PY14Br), to synthesize hydrate molten electrolytes. This approach effectively reduces the activity of H₂O within the electrolyte by facilitating strong ion associations between water molecules and loosely bound halogen anions, ultimately enhancing the electrochemical stability of the battery. This work is in general of novelty for the AAB field. However, some important issues have not be clearly clarified, which require further investigation for understanding the effect of the additive. This manuscript can be further considered after addressing those issues.

Response: We appreciate the reviewer's positive comment and valuable suggestions. Additional experiments and detailed discussions have been conducted to address the following concerns:

1. The authors have demonstrated through ab initio molecular dynamics (AIMD) simulations that over 98% of water molecules in the electrolyte contribute to the formation of the Al solvation structure. I would like to inquire whether the authors have validated the accuracy of the AIMD simulations through additional phase characterizations, such as spectral tests.

Fig. R24 Experimental and AIMD-simulated WAXS spectra of **a** 3.25 m AlCl₃, **b** 3.25 m AlCl₃ + 0.5 m KBr, and **c** 3.25 m AlCl₃ + 1 m PY14Br.

Response: Thank you for the valuable comment. To validate the accuracy of our AIMD simulations, we derived wide-angle X-ray scattering (WAXS) spectra from the simulated results through Debye's scattering equation.¹⁶ These simulated WAXS spectra were then compared with our experimental WAXS data. As shown in **Fig. R24**, the simulated spectra exhibit excellent alignment with the experimental spectra, demonstrating nearly identical spectral features. This strong agreement validates the reliability and accuracy of the AIMD simulation results.

Action: The relevant discussion has been added to the revised manuscript. (Page 8, Paragraph 2)

2. What is the molecular structure of the Al solvation complex that fully incorporates water molecules in the mixed electrolyte? Additionally, could you provide the corresponding experimental characterization?

Response: We appreciate the insightful question of the reviewer. To illustrate the Al solvation complex, we derived a typical structure from the AIMD simulation result of 3.25 m AlCl₃ + 1 m PY14Br (Fig. R25a). The complex contains two solvation sheaths which are dominated by water molecules, and majority of halogen anions (Cl⁻/Br⁻) are located at the outer solvation sheath of Al, as determined by the radial distribution function (RDF) analysis (Fig. 2e-g and Supplementary Fig. 13-15). As explained in our response to Q1, the reliability of the simulated Al solvation complex structure can be confirmed by the good agreement between the simulated and experimental WAXS spectra. Moreover, the simulation result is also in consistent with the ²⁷Al nuclear magnetic resonance (NMR) spectrum of the electrolyte (Fig. R25b), in which a dominant peak at 0 ppm was detected, corresponding to Al³⁺ with water molecules associated in the inner solvation sheath (i.e., Al(H₂O)₆³⁺).¹⁷

Action: We have included the corresponding discussion in the revised manuscript. (Page 7, Paragraph 3 and Page 8, Paragraph 1-3)

Fig. R25 a Representative Al³⁺ solvation structure extracted from the snapshot of an AIMD simulated cell for 3.25 m AlCl₃ + 1 m PY14Br. **b** ²⁷Al NMR spectra of 3.25 m AlCl₃, 3.25 m AlCl₃ + 0.5 m KBr, 3.25 m AlCl₃ + 1 m PY14Br, and 3.25 m AlCl₃ + 1 m Et₄NBr.

3. To emphasize the structural advantages of PY14Br, it is recommended to include the results of the ²⁷Al nuclear magnetic resonance test for the AlCl₃ + EtN₄Br system.

Response: According to your valuable suggestion, we collected the ^{27}Al NMR spectrum of 3.25 m AlCl_3 + 1 m Et_4NBr and compared it with the spectra of 3.25 m AlCl_3 , 3.25 m AlCl_3 + 0.5 m KBr , and 3.25 m AlCl_3 + 1 m PY14Br (**Fig. R25b**). Similar to the other three electrolytes, 3.25 m AlCl_3 + 1 m Et_4NBr presents only a dominant peak at 0 ppm, which corresponds to Al^{3+} coordinated with six water molecules in the inner solvation sheath (i.e., $\text{Al}(\text{H}_2\text{O})_6^{3+}$). This observation indicates neither PY14Br nor Et_4NBr significantly alters the inner solvation structure of Al^{3+} .

Action: The relevant discussion has been added to the revised manuscript. (**Supplementary Fig. 26**)

4. Does the incorporation of organic PY14Br into the mixed electrolyte lead to a reduction in conductivity? Furthermore, given the importance of electrolyte additive stability for battery cycle longevity, it is recommended that the authors also assess the electrochemical stability of PY14Br throughout cycling.

Response: Thank you for the insightful questions. Accordingly, we measured the ionic conductivity of our electrolytes by electrochemical impedance spectroscopy (EIS) at 298.15 K (**Fig. R26a**). Specifically, a sealed 2-electrode cell consisting of two platinum electrodes placed parallel to one another with a nominal constant (C) of 1.021 cm^{-1} was used. The EIS measurements were carried out at a 20 mV AC oscillation amplitude over the frequency range of 100 kHz to 0.01 Hz. The intercept along the real axis intrinsic shows the resistance of the electrolyte (R). Ionic conductivities were calculated according to equation (**R5**). **Table R4** lists the calculated ionic conductivity values. It was revealed that the addition of PY14Br reduced the ionic conductivity from 52.9 mS cm^{-1} for 3.25 m AlCl_3 to 32.9 mS cm^{-1} for 3.25 m AlCl_3 + 1 m PY14Br . This reduction in ionic conductivity can be attributed to the electrolyte viscosity increase, as measured by rotational viscometer (38 mPa s for 3.25 m AlCl_3 and 58 mPa s for 3.25 m AlCl_3 + 1 m PY14Br). The ionic conductivity values are larger than that of the classic highly concentrated electrolytes, such as 21 m LiTFSI (8.2 mS cm^{-1} at 298.15 K) and 1 m LiTFSI + 7 m LiBETI (3.0 mS cm^{-1} at 303.15 K).^{2,5}

$$\sigma = \frac{C}{R} \text{ (R5)}$$

Table R4 Viscosity and ionic conductivity of different electrolytes.

Electrolytes	Ionic conductivity (mS cm^{-1})	Viscosity (mPa s)
3.25 m AlCl_3	52.9	38
3.25 m AlCl_3 + 0.5 m KBr	59.3	42

3.25 m AlCl ₃ + 1 m PY14Br	32.9	58
3.25 m AlCl ₃ + 1 m Et ₄ NBr	36.2	56

Regarding the electrochemical stability of PY14Br, we collected the 3.25 m AlCl₃ + 1 m PY14Br electrolyte from an Al-Br cell after 10 charge/discharge cycles at 5 mA cm⁻² and 5 mAh cm⁻². This cycled electrolyte was then analyzed via Fourier-transform infrared spectroscopy (FT-IR) and compared with the original 3.25 m AlCl₃ and 3.25 m AlCl₃ + 1 m PY14Br electrolytes. As shown in **Fig. R26b**, the cycled 3.25 m AlCl₃ + 1 m PY14Br electrolyte displays a nearly identical FT-IR spectrum to the original one. Notably, the characteristic PY14⁺ peak at 1467 cm⁻¹ remains clearly detectable, confirming the superior electrochemical stability of PY14Br. In fact, PY14⁺ is well-known for its high electrochemical stability, as evidenced by its successful use in electrolytes for non-aqueous battery systems, which typically operate over significantly broader potential windows than the one used in our system.^{18,19}

Fig. R26 a EIS analysis for 3.25 m AlCl₃, 3.25 m AlCl₃ + 0.5 m KBr, 3.25 m AlCl₃ + 1 m PY14Br, and 3.25 m AlCl₃ + 1 m Et₄NBr. **b** FT-IR spectra of 3.25 m AlCl₃, 3.25 m AlCl₃ + 1 m PY14Br, and the cycled 3.25 m AlCl₃ + 1 m PY14Br electrolyte.

Action: The corresponding discussion has been added to the revised manuscript. (Page 7, Paragraph 2 and Page 15, Paragraph 1)

5. The energy storage device assembled by the authors employs an Al/Zn alloy as the negative electrode. Have the authors investigated whether Zn participates in the electrochemical reaction and, if so, its contribution to capacity?

Response: We appreciate the valuable question from the reviewer. To assess the participation of Zn, we first conducted a metal stripping experiment for the Zn-Al alloy electrode in 3.25 m AlCl₃ + 1 m PY14Br at 5 mA cm⁻² for 1 h, using a three-electrode setup comprising a Zn-Al alloy working electrode with an area of 1 cm², an over-capacity AC electrode as the counter electrode, and an Ag/AgCl reference electrode. Afterward, the electrolyte was extracted from the cell and analyzed via inductively coupled plasma optical emission spectroscopy (ICP-OES) to quantify the stripped Zn in the electrolyte. The stripped Zn was determined to be 3.6×10^{-6} mol, corresponding to a capacity of 0.195 mAh. Based on this analysis, we conclude that the participation of Zn in the process is minimal, accounting for only 3.9 % of the total Zn-Al alloy capacity.

Action: We have included the relevant discussion in the revised manuscript. (Page 14, Paragraph 3)

6. On page 4, line 82, what does 3.25 m mean? Is it mol L⁻¹?

Response: The unit 'm' is defined as the moles of a solute per kilogram of a solvent, commonly used for highly concentrated electrolytes.

Action: We have clarified it in the revised manuscript. (Page 4, Paragraph 2)

References:

- 1 Yang, M., Zhu, J., Bi, S., Wang, R. & Niu, Z. A binary hydrate-melt electrolyte with acetate-oriented cross-linking solvation shells for stable zinc anodes. *Adv. Mater.* **34**, 2201744 (2022).
- 2 Yamada, Y. *et al.* Hydrate-melt electrolytes for high-energy-density aqueous batteries. *Nat. Energy* **1**, 16129 (2016).
- 3 Chen, X. *et al.* Raman spectroscopic investigation of tetraethylammonium polybromides. *Inorg. Chem.* **49**, 8684-8689 (2010).
- 4 Xu, C. *et al.* Practical high-energy aqueous zinc-bromine static batteries enabled by synergistic exclusion-complexation chemistry. *Joule* **8**, 461-481 (2024).
- 5 Suo, L. *et al.* "Water-in-salt" electrolyte enables high-voltage aqueous lithium-ion chemistries. *Science* **350**, 938-943 (2015).
- 6 Wen, B. *et al.* Concentrated chlorine-based electrolyte enabling reversible $\text{Cl}_3^-/\text{Cl}^-$ redox for energy-dense and durable aqueous batteries. *ACS Energy Lett.* **8**, 4204-4209 (2023).
- 7 Cai, S. *et al.* Water-salt oligomers enable supersoluble electrolytes for high-performance aqueous batteries. *Adv. Mater.* **33**, 2007470 (2021).
- 8 Shi, J. *et al.* In situ p-block protective layer plating in carbonate-based electrolytes enables stable cell cycling in anode-free lithium batteries. *Nat. Mater.* **23**, 1686-1694 (2024).
- 9 Dai, C. *et al.* Fast constructing polarity-switchable zinc-bromine microbatteries with high areal energy density. *Sci. Adv.* **8**, eabo6688
- 10 Wang, C. *et al.* Visualizing and understanding the ionic liquid-mediated polybromide electrochemistry for aqueous zinc-bromine redox batteries. *Nano Lett.* **24**, 13796-13804 (2024).
- 11 Li, X. *et al.* A complexing agent to enable a wide-temperature range bromine-based flow battery for stationary energy storage. *Adv. Funct. Mater.* **31**, 2100133 (2021).
- 12 Chen, Y., Zhao, B., Yang, Y. & Cao, A. Toward high-areal-capacity electrodes for lithium and sodium ion batteries. *Adv. Energy Mater.* **12**, 2201834 (2022).
- 13 Hao, J. *et al.* Deeply understanding the Zn anode behaviour and corresponding improvement strategies in different aqueous Zn-based batteries. *Energy Environ. Sci.* **13**, 3917-3949 (2020).
- 14 Mahmood, A., Zheng, Z. & Chen, Y. Zinc-bromine batteries: Challenges, prospective solutions, and future. *Adv. Sci.* **11**, 2305561 (2024).
- 15 Li, Q., Chen, A., Wang, D., Pei, Z. & Zhi, C. "Soft shorts" hidden in zinc metal anode research. *Joule* **6**, 273-279 (2022).
- 16 Debye, P. Zerstreung von Röntgenstrahlen. *Ann. Phys.* **351**, 809-823 (1915).
- 17 Maki, H., Sakata, G. & Mizuhata, M. Quantitative NMR of quadrupolar nucleus as a novel analytical method: hydrolysis behaviour analysis of aluminum ion. *Analyst* **142**, 1790-1799 (2017).
- 18 Zhou, W., Zhang, M., Kong, X., Huang, W. & Zhang, Q. Recent advance in ionic-liquid-based electrolytes for rechargeable metal-ion batteries. *Adv. Sci.* **8**, 2004490 (2021).
- 19 Elia, G. A., Ulissi, U., Jeong, S., Passerini, S. & Hassoun, J. Exceptional long-life performance of lithium-ion batteries using ionic liquid-based electrolytes. *Energy Environ. Sci.* **9**, 3210-3220 (2016).

To Reviewer 2:

The reply content meets most of the research problems, but the discussion on the selection principles of bromine salts is insufficient, it is necessary to provide a definitive data-oriented summary to explain the essential issue on the bromine salt in the term of different organic cations. And I would recommend the publication after further revision of the review.

Response: We sincerely appreciate the valuable comments and the recommendation for publication. Following your suggestion, we have added further discussion on the selection principle for bromine salts, especially focusing on the role of organic cations.

Fig. R1 Digital photos of **a** 3.25 m AlCl_3 + 1 m PY14Br and **b** 3.25 m AlCl_3 + 0.5 m KBr after the galvanostatic charge test at 1 mA cm^{-2} and 5 mAh cm^{-2} using a three-electrode setup with a glass beaker.

Our early results indicate that organic cations can electrostatically associate with anionic polyhalide species (e.g., Br_{2n+1}^- and BrCl_2^-), forming water-insoluble phases (**Fig. R1a**). These water-insoluble phases help suppress the dissolution of redox-active species into the aqueous electrolyte, thereby improving Coulombic efficiency. In contrast, inorganic cations such as K^+ are ineffective at stabilizing these polyhalide species (**Fig. R1b**). Consequently, an organic cation with a stronger binding affinity toward anionic polyhalide species than toward water molecules is more likely to promote the formation of stable, water-insoluble phases rather than facilitating their dissolution. To derive data-driven insights, we examined four available bromide salts with bulky organic cations: PY14Br, Et_4NBr , triethylamine hydrobromide (Et_3NHBr), and tetramethylammonium bromide (Me_4NBr). We calculated the binding energies of their cations with Br_3^- , BrCl_2^- , and H_2O (**Table R1**). All organic cations exhibited stronger bindings with $\text{Br}_3^-/\text{BrCl}_2^-$ than with H_2O , indicating their tendency to form water-insoluble phases, consistent with experimental observations. To quantify this trend, we defined two parameters: the binding energy difference of the organic cation between with Br_3^- and with H_2O (E_1), the binding energy

difference of the organic cation between with BrCl_2^- and with H_2O (E_2). These parameters reflect the relative affinity of the cations toward polyhalide species compared toward water. As shown in **Fig. R2a**, both E_1 and E_2 follow the order: $\text{PY14}^+ > \text{Et}_4\text{N}^+ > \text{Et}_3\text{NH}^+ > \text{Me}_4\text{N}^+$, with the trend in E_1 being more pronounced. This trend is further corroborated by the GCD measurements (**Fig. R2b**), where the Coulombic efficiencies of the AC electrodes in the three-electrode test follow the same order (**Fig. R2c**): $3.25 \text{ m AlCl}_3 + 1 \text{ m PY14Br}$ (99.6%) $>$ $3.25 \text{ m AlCl}_3 + 1 \text{ m Et}_4\text{NBr}$ (98.6%) $>$ $3.25 \text{ m AlCl}_3 + 1 \text{ m Et}_3\text{NHBr}$ (97.7%) $>$ $3.25 \text{ m AlCl}_3 + 1 \text{ m Me}_4\text{NBr}$ (96.2%). These findings highlight E_1 and E_2 as effective descriptors for selecting suitable bromide salts.

Fig. R2 a Calculated binding energy differences of various organic cations with Br_3^- and BrCl_2^- relative to with H_2O . **b** GCD profiles of AC electrodes in different electrolytes at 5 mA cm^{-2} and 5 mAh cm^{-2} . **c** Coulombic efficiencies of the AC electrodes using different electrolytes in the three-electrode test.

Table R1 Binding energies of organic cations with Br_3^- , BrCl_2^- , and H_2O .

Organic cation	Binding energy with Br_3^- (eV)	Binding energy with BrCl_2^- (eV)	Binding energy with H_2O (eV)
PY14 ⁺	-4.17423	-3.49277	-0.48032
Et ₄ N ⁺	-4.07061	-3.44136	-0.43592
Et ₃ NH ⁺	-3.95720	-3.44645	-0.46534
Me ₄ N ⁺	-3.85106	-3.51762	-0.53792

Action: We have included the relevant discussion into the revised manuscript. (Page 10, Paragraph 1)

To Reviewer 3:

The authors have revised this paper carefully based on the reviewer's comments. I'd like to recommend its publication without alternation.

Response: We sincerely appreciate your positive feedback and recommendation for publication.

To Reviewer 4:

The work titled “Hydrate-Melt Electrolyte Design for Aqueous Aluminium-1 Bromine Batteries with Superior Energy-Power Merits” presents the fabrication of AlCl_3 -based aqueous hydrate-melt electrolyte, which demonstrates good compatibility with both Zn/Al anodes and the $\text{Br}^-/\text{Br}_0/\text{Br}^+$ conversion. The unique double-sheath solvation structure of Al^{3+} revealed in the electrolyte enables both low water reactivity and a wide electrochemical stability window. The proposed electrolyte system is well characterized through a combination of coherent experimental and computational investigations. When using this electrolyte in Al-Br batteries, the designed electrolytes are endowed with high power density and high areal capacity up to 5 mAh cm^{-2} . In general, this manuscript is well organized, the high areal capacity for the cathode has demonstrated potential application interest. I have shown that in previous round of peer-review process, the three reviewers have raised insightful comments to improve the scientificity of manuscript, which have been mostly addressed by the authors. Therefore, in my opinion, the manuscript justifies publication in Nature Communications. I only have a few technical points needing further clarification by the authors.

Response: We appreciate the positive comments and valuable suggestions of the reviewer. Additional experiments and detailed discussions have been conducted to address the following concerns:

1. The authors have attributed the great rate performance of the Al-Br battery to fast redox kinetics. To strengthen this claim, it would be helpful to include electrochemical impedance spectroscopy measurements to directly evaluate charge transfer characteristics. In addition, reporting the energy efficiency of the full cell at various current densities would provide valuable insight into the practical performance of the system under different operating conditions.

Response: Thank you for the valuable suggestion. Accordingly, we conducted electrochemical impedance spectroscopy (EIS) measurements on the AC electrode at different charge states (0%, 50%, and 100%) in $3.25 \text{ m AlCl}_3 + 1 \text{ m PY14Br}$. As shown in **Fig. R3a**, the AC electrode maintains a low ohmic resistance (below 0.5Ω) and charge transfer resistance (below 10Ω) across all charge states. This observation indicates the fast kinetics of the Br conversion within the AC electrode. Additionally, we evaluated the energy efficiencies of the Al-Br full cell at various current densities (**Fig. R3b**). At a current density of 5 mA cm^{-2} (1 C), the full cell achieves a high energy efficiency of 84%. Even at 20 mA cm^{-2} (4 C), the energy efficiency remains as high as 78%, underscoring the excellent rate capability and practical viability of the Al-Br cell.

Action: The relevant discussion has been added to the revised manuscript. (**Page 11, Paragraph 2 and Page 15, Paragraph 3**)

Fig. R3 **a** Nyquist plots of the AC electrode at different charge states (0%, 50%, and 100%). **b** Energy efficiencies of the Al-Br full cell at different current densities.

2. In practical applications, the operating temperature is not constant. Since the proposed hydrate-melt electrolyte relies on a highly structured solvation environment, it would be valuable to comment on whether this structure remains stable at different temperatures. If possible, the authors are encouraged to explore this aspect using AIMD or complementary experimental data.

Response: We appreciate the insightful comment of the reviewer. Based on our early results (**Supplementary Fig. 44**), the cathode performance remains stable across a temperature range from room temperature (298.15 K) to 333.15 K. Following the reviewer's suggestion, we further conducted AIMD simulations of 3.25 m AlCl_3 + 1 m PY14Br at 313.15 K and 333.15 K to assess the influence of temperature on the electrolyte structure. As reflected by the derived radial distribution functions (RDFs) of Al atoms (**Fig. R4**), the electrolyte maintains a stable structural configuration at elevated temperatures, comparable to that at 298.15 K. Specifically, the majority of water molecules engage in robust ion solvation, while halogen anions reside primarily in the outer solvation sheath of Al^{3+} . These results confirm that the highly structured solvation environment is well preserved over a wide temperature range, consistent with the experimentally observed robust electrochemical performance.

Action: The relevant discussion has been added to the revised manuscript. (**Page 17, Paragraph 2**)

Fig. R4 RDFs (solid lines) and integral curves (dashed lines) of Al atoms extracted from AIMD simulations of 3.25 m AlCl_3 + 1 m PY14Br at different temperatures, **a** Al–O_{water}, **b** Al–Cl, and **c** Al–Br at room temperature (297.15 K), **d** Al–O_{water}, **e** Al–Cl, and **f** Al–Br at 313.15 K, as well as **h** Al–O_{water}, **i** Al–Cl, and **j** Al–Br at 333.15 K.

3. The manuscript reports impressive capacity values enabled by the $\text{Br}^-/\text{Br}^0/\text{Br}^+$ conversion chemistry within an activated carbon host. To fully understand the contribution of the redox species, it would be helpful to clarify the intrinsic capacity of the activated carbon itself. Providing a control experiment to estimate baseline capacity from the carbon host alone would help quantify the actual contribution from the Br-based redox process.

Response: Thank you for the constructive suggestion. To evaluate the capacity contribution from the activated carbon (AC) host, we conducted GCD measurements of the AC electrode in 3.25 m AlCl_3 to exclude the capacity contribution from Br conversion. As shown in **Fig. R5**, the capacitive charge storage of activated carbon accounts for only 0.28 mAh cm^{-2} , far below the total capacity of the AC

electrode in 3.25 m AlCl₃ + 1 m PY14Br (4.97 mAh cm⁻²). This result clearly confirms that the predominant capacity arises from the Br⁻/Br⁰/Br⁺ conversion.

Action: The relevant discussion has been added to the revised manuscript. (Page 7, Paragraph 2)

Fig. R5 GCD profiles of the AC electrodes at 5 mA cm⁻² in 3.25 m AlCl₃ and 3.25 m AlCl₃ + 1 m PY14Br.

4. The electrolyte exhibits excellent anodic stability, thereby enabling the reversible Br⁻/Br⁰/Br⁺ conversion chemistry. In the manuscript, the anodic stability of the baseline electrolyte (3.25 m AlCl₃) is confirmed to extend beyond 1.5 V vs. Ag/AgCl. The authors then introduce the Br source in the form of organic bromide salts. It would be important to clarify whether the introduced organic cation affects the anodic stability. The authors are encouraged to perform LSV or potential floating measurements using 3.25 m AlCl₃ + 1 m PY14Cl to evaluate the influence of the organic cation.

Response: To address the reviewer's concern, we performed the LSV measurement in 3.25 m AlCl₃ + 1 m PY14Cl to evaluate the effect of the introduced organic cations on the electrolyte anodic stability. As indicated by **Fig. R6**, 3.25 m AlCl₃ + 1 m PY14Cl electrolyte only exhibits slightly reduced anodic stability compared to 3.25 m AlCl₃, yet significantly better than 0.1 m AlCl₃ and 1 m AlCl₃. Importantly, it shows a negligible anodic current density (1.2×10^{-4} mA cm⁻²) at 1.5 V vs. Ag/AgCl, a potential well above the Br⁻/Br⁰/Br⁺ conversion range (0.6 ~ 0.7 V vs. Ag/AgCl for Br⁻/Br⁰ and 0.9 ~ 1.0 V vs. Ag/AgCl for Br⁰/Br⁺). These findings confirm that the introduction of organic cations does not compromise the anodic stability necessary for reversible Br conversion.

Fig. R6 Linear sweep voltammetry curves of the Ti electrode in varying electrolytes at a scan rate of 10 mV s^{-1} .

Action: The relevant discussion has been added to the revised manuscript. (**Page 7, Paragraph 2**)

5. The authors describe the proposed electrolyte as “cost-effective”, which is a critical consideration for practical applications, particularly in large-scale energy storage. This claim would be more convincing if it is supported by comparative data. Therefore, it is recommended that the authors compare the cost of this electrolyte with other representative aqueous Al battery electrolytes, such as $\text{Al}(\text{OTF})_3$, of course taking consideration of salt concentration.

Response: We appreciate the valuable suggestion from the reviewer. To substantiate the cost-effectiveness of the proposed electrolyte, we conducted a comparative cost analysis between our proposed electrolytes and representative $\text{Al}(\text{OTF})_3$ -based electrolytes commonly reported for aqueous Al battery.¹⁻³ All pricing data are obtained from Sigma-Aldrich, a standard supplier of laboratory-grade chemicals. As summarized in **Table R2**, the salts employed in this work ($\text{AlCl}_3 \cdot 6\text{H}_2\text{O}$, PY14Br, and Et_4NBr) are significantly less expensive than $\text{Al}(\text{OTF})_3$. We further estimated the total cost of the prepared electrolytes (**Table R3**). Both electrolytes used in this work, $3.25 \text{ m AlCl}_3 + 1 \text{ m PY14Br}$ and $3.25 \text{ m AlCl}_3 + 1 \text{ m Et}_4\text{NBr}$, demonstrate substantially lower costs compared to $\text{Al}(\text{OTF})_3$ -based electrolytes at various concentrations reported previously for aqueous Al batteries. These findings confirm that our electrolytes, particularly the Et_4NBr -based formulation, present a more economically viable option for large-scale energy storage applications.

Action: The relevant discussion has been added to the revised manuscript. (Page 7, Paragraph 2)

Table R2 Cost comparison of key salts used in aqueous Al battery electrolytes

Salts	Price (€/kg)	Price (€/mol)	Supplier
$\text{AlCl}_3 \cdot 6\text{H}_2\text{O}$	200	48.3	Sigma-Aldrich
PY14Br	4260	946.4	Sigma-Aldrich
Et_4NBr	196	41.2	Sigma-Aldrich
$\text{Al}(\text{OTf})_3$	8380	3973.7	Sigma-Aldrich

Table R3 Estimated cost of aqueous Al battery electrolytes.

Electrolytes	Price (€/kg)
3.25 m AlCl_3 + 1 m PY14Br	666.7
3.25 m AlCl_3 + 1 m Et_4NBr	113.3
2 m $\text{Al}(\text{OTf})_3$	4079.8
5 m $\text{Al}(\text{OTf})_3$	5895.7

References

- 1 Wu, C. *et al.* Electrochemically activated spinel manganese oxide for rechargeable aqueous aluminum battery. *Nat. Commun.* **10**, 73 (2019).
- 2 Zhao, Q. *et al.* Solid electrolyte interphases for high-energy aqueous aluminum electrochemical cells. *Sci. Adv.* **4**, eaau8131
- 3 Ran, Q. *et al.* Aluminum-copper alloy anode materials for high-energy aqueous aluminum batteries. *Nat. Commun.* **13**, 576 (2022).